# Lignocellulose-mediated selection of potential halophilic PET-degrading enzymes from mangrove soil

María Fernanda Peña-Valencia[1,11], Semidán Robaina-Estévez [2], Gordon F. Custer[3], Onur Turak[4], Felipe Sierra [1], Lucas William Mendes [5], Carolina Rubiano-Labrador[6], Jay Gutiérrez[2], Annika Vaksmaa[7], Francisco Dini-Andreote [8,9], Alexandre Soares Rosado [10] ✉, Alejandro Reyes [1] ✉ & Diego Javier Jiménez [10,11] ✉

Mangroves are ecosystems located at land–sea transition zones, where they are continuously exposed to plant biomass and plastic pollution. Their soils harbor extensive microbial diversity with potential for discovering polymer-degrading enzymes. Here, we perform a microcosm experiment to examine how mangrove soil microbial communities respond to inputs of lignocellulose or polyethylene terephthalate (PET) in the presence and absence of seawater, and to explore the selection of putative PET-active enzymes (PETases) using gene- and genome-resolved metagenomics. Incubation conditions lead to a gradual increase in salinity, resulting in the enrichment of halophilic taxa, including spore-forming bacteria and archaeal species, particularly in seawater-depleted treatments. Lignocellulose input is the primary driver of soil microbial community restructuring, followed by seawater presence. In dry, lignocellulose-amended microcosms (L treatment), microbial diversity is significantly reduced, while lignocellulolytic taxa within the phyla *Bacillota* and *Actinomycetota* are enriched. Twelve potential PETases are identified in the L treatment, sharing >70% sequence similarity with known PETases, and three are predicted to be thermostable. Two putative PETases from *Microbulbifer* species display distinct sequence and structural features, thereby expanding the currently limited PETase sequence landscape. This study demonstrates that perturbing environmental microbiomes with plant-derived polymers represents a promising strategy for capturing novel PETases.

Mangroves are critical ecosystems that provide multiple global services, including carbon sequestration, climate change mitigation, and biodiversity support[1,2]. Located at the terrestrial-marine interface, these ecosystems are dynamic and shaped by tidal currents, hosting diverse microbial species that participate in complex interactions, metabolic processes, and ecological functions[3–5]. The microbial communities inhabiting mangrove soils are predominantly composed of halotolerant species dispersed through seawater floods[6–8]. Mangrove soils are prone to multiple environmental changes and disturbances. For example, (i) constant flooding introduces new microbial species and promotes nutrient runoff[9,10]; (ii) pollutant inputs (e.g., microplastics and agricultural residues) can modify

physicochemical conditions and alter microbial community structures[11,12]; and (iii) drought events increase soil salinity, which reduces plant productivity and elevates mortality rates[13,14].

Globally, plastic pollution represents a major environmental challenge, with polyethylene terephthalate (PET) being a significant contributor to marine waste. Mangrove ecosystems, as natural sinks, are particularly vulnerable to the accumulation of fossil-derived plastics[15,16]. Exposure of mangrove soil microbiomes to tidal flooding, salinity shifts, plant-derived polymers[17,18], and microplastics[19,20] makes these ecosystems promising reservoirs of novel lignocellulose- and plastic-transforming enzymes with polyextremophilic traits potentially suitable for industrial applications[21,22]. Halophilic enzymes, such as lipases and (hemi)cellulases, are valuable for numerous biotechnological applications, including their use as active compounds in detergents and as components of enzymatic cocktails for the saccharification of agricultural residues in biorefineries[7,23]. However, halophilic PET-depolymerizing enzymes from mangrove-associated environments have not yet been described. Recently, we reported five putative PET-depolymerizing enzymes from *Pseudoxanthomonas winnipegensis*, an abundant member of PET-transforming bacterial consortia selected from a mangrove soil microcosm[24].

PET-depolymerizing enzymes are α/β-fold hydrolases (known as PETases), primarily found in taxa within the phyla *Pseudomonadota*, *Actinomycetota*, and *Bacillota*[25,26]. Based on amino acid sequences and 3D structural conformations, these enzymes have been classified into three main groups (I, IIa, and IIb)[27]. More recently, a fourth group of marine-derived PETases (group III) has been proposed, encompassing salt-tolerant enzymes from the *Halopseudomonas* lineage[28]. Additionally, three halophilic PETases with high homology to IsPETase (from *Piscinibacter sakaiensis*, formerly *Ideonella sakaiensis*)[29], have been identified in protein catalogs from deep-sea sediment metagenomes[30,31]. From an industrial perspective, PETases that remain active under extreme conditions, such as high temperatures (~70 °C), low pH, highly crystalline PET, and elevated salt concentrations, are particularly desirable for optimizing biocatalytic processes[26,31–33].

Ecological perturbation experiments enable the investigation of dynamic responses and resilience of microbial communities under acute or chronic physicochemical changes[34]. However, these experiments with soil microbiomes (e.g., microcosms) can be adapted to select for target or rare microbial populations[35] with PET-depolymerizing capabilities[24,36]. For example, a recent two-year microcosm experiment using deep-sea sediments and PET microparticles successfully selected PET-degrading prokaryotic species[37]. Although PET microparticles or films are rational inducers of such selection, other easy-to-break PET analog substrates (e.g., plant biomass or polyester-rich substrates) can also serve as effective disturbance factors, increasing the likelihood of enriching PET-depolymerizing taxa and enzymes[38]. This is because natural substrates of PETases, such as plant polymers (e.g., xylan, suberin, and cutin), contain the same ester bonds present in PET[39,40]. Therefore, selecting for lignocellulose-degrading microbial communities in soil microcosms represents a promising strategy for enriching prospective PET-active enzymes. This idea is further supported by the frequent occurrence of polyester-degrading enzymes in plant polymer-rich environments such as compost, soils, and herbivore gastrointestinal tracts[40,41].

In this work, we design a microcosm experiment to (i) evaluate the responses of mangrove soil microbial communities (i.e., shifts in diversity and structure) to inputs of lignocellulosic substrate or amorphous PET microparticles in a saline environment with or without seawater, and (ii) assess the selection of polymer-degrading prokaryotic taxa and enzymes. By reconstructing metagenome-assembled genomes (MAGs) and generating comprehensive gene catalogs, we screen the enzymatic potential of these restructured soil microbial communities for the transformation of plant biomass and PET. In this study, we demonstrate that ex situ perturbation of mangrove-derived soils with lignocellulose substantially reshapes microbial communities, enriching halophilic species with the capacity to transform plant polymers. The inclusion of lignocellulose in mangrove soil microcosms is particularly associated with the selection of prokaryotic lineages that harbor an expanded genetic potential for PET depolymerization, including putative halophilic PETases.

## Results

### Reshaping of mangrove soil microbial communities in lignocellulose-amended microcosms

A soil microcosm experiment was conducted with lignocellulose (rice husk) and PET particles amendments, along with two environmental factors: (i) the presence of external seawater-derived microbial populations and nutrients, and (ii) a gradual increase in soil desiccation and salinity over the incubation period (30 and 90 days) at 30 °C (Fig. 1). Based on Shannon index values derived from 16S rRNA gene and ITS2 amplicon sequencing, microcosms containing lignocellulose without seawater (L treatment) exhibited a trend toward lower alpha diversity compared to other treatments without seawater after 30 days of incubation (ANOVA; $p > 0.08$, Fig. 2A). A significant pairwise difference in bacterial Shannon diversity was observed between the L treatment and controls with seawater (CW), with the latter showing higher diversity at 30 days ($p < 0.05$). After 90 days, bacterial alpha diversity decreased significantly in microcosms without seawater compared to those with seawater inclusion ($p < 0.001$, Fig. 2A). Fungal alpha diversity in the L treatment displayed a distinctive pattern, with an initial decline at 30 days followed by an increase at 90 days, suggesting recovery and re-assembly of the fungal community after perturbation.

Beta diversity analysis based on Bray-Curtis distances revealed that both bacterial and fungal communities were significantly reshaped by lignocellulose input (PERMANOVA; $p < 0.001$, $R^2 = 0.187$ and 0.199), with seawater inclusion also contributing significantly to community variation ($p < 0.001$, $R^2 = 0.122$ and 0.102, Fig. 2B). These compositional changes were reflected in the relative abundance of dominant phyla after 90 days, including *Bacillota* (in L and LW), *Bacteroidota* (in LW), and *Planctomycetota* (in L) (ANOVA; $p < 0.001$, $R^2 = 0.057$, Fig. 2C). Furthermore, combined bacterial and fungal co-occurrence network analysis revealed that lignocellulose input substantially reduced the complexity of the soil microbial community after 90 days, as evidenced by a decrease in the number of nodes and edges compared with negative controls (C and CW) and the original mangrove soil microbial community (S) (Supplementary Data 1 and Fig. 2D).

### Enrichment of spore-forming bacterial taxa in lignocellulose-amended microcosms

The Boruta algorithm identified 102 bacterial ASVs associated with either a specific treatment group or the original mangrove soil bacterial community (S) (Fig. 3A). Among these, taxa within the families *Bacillaceae*, *Sporolactobacillaceae*, and *Paenibacillaceae* showed higher relative abundances in lignocellulose-amended microcosms (L and LW) after 90 days of incubation. Most of these ASVs were enriched in the lignocellulose treatments and exhibited lower abundances in PET-amended microcosms (P and PW), negative controls (C and CW), and in the original soil bacterial community (S) (Fig. 3A). Differential abundance analysis using DESeq2 revealed a significant enrichment ($p < 0.0001$; log2-fold change > 5) of 86 ASVs in lignocellulose-amended microcosms compared with negative controls (Fig. 3B). Notably, some ASVs assigned to *Bacillaceae* (3 ASVs), *Micromonosporaceae* (1 ASV), *Paenibacillaceae* (1 ASV), and *Pirellulaceae* (1 ASV) displayed particularly high relative abundances, with fold-changes greater than 9 in lignocellulose-amended treatments over the 30–90 day incubation period.

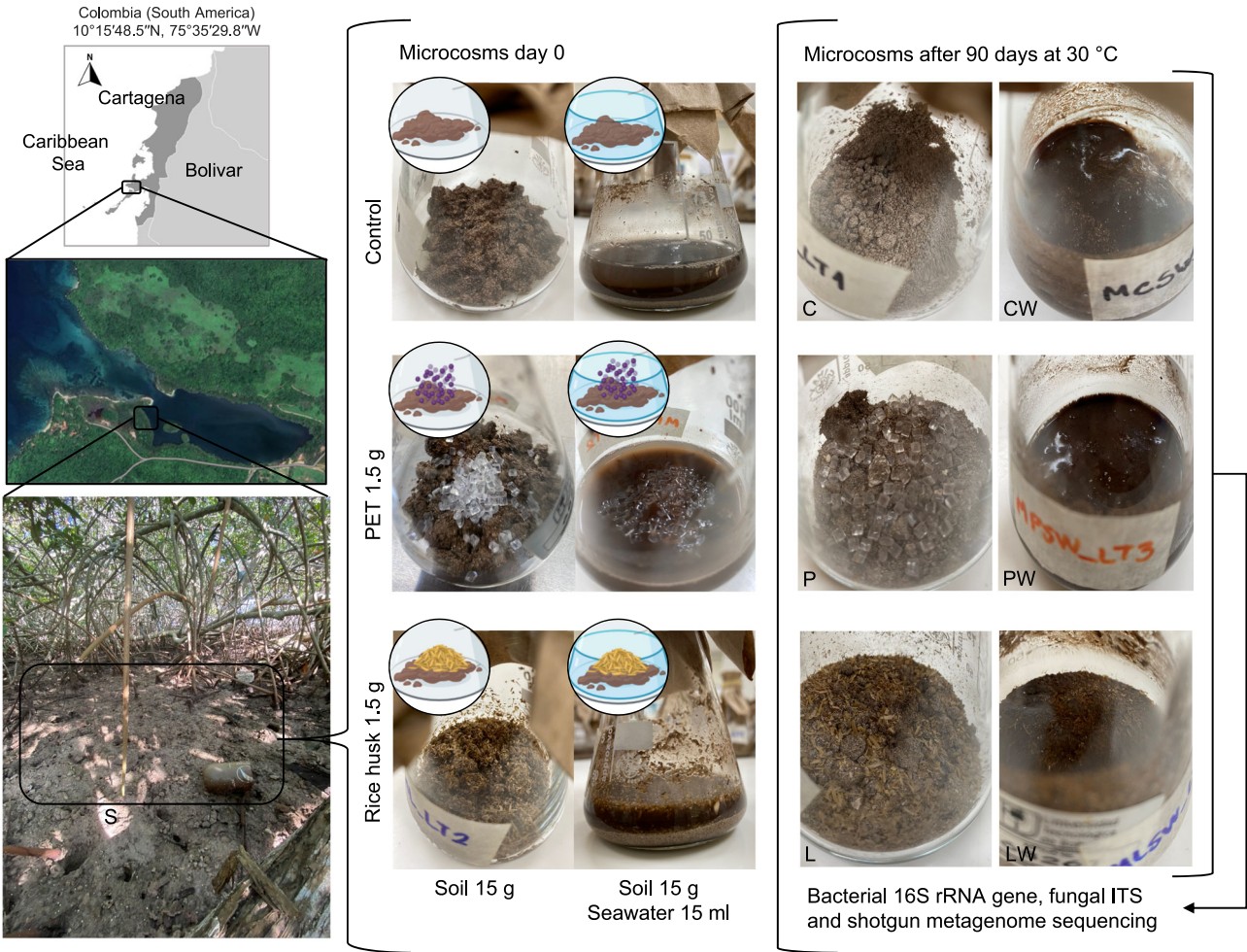

**Fig. 1 | Experimental design and microcosm development.** The left panel shows the site where soil samples were collected. The central panel depicts the microcosm experimental setup, which included six treatments: control without water (C), control with water (CW), PET without water (P), PET with water (PW), rice husk without water (L), and rice husk with water (LW). All microcosms were incubated at 30 °C under static conditions for 30 and 90 days. The photographs illustrate the treatments before incubation (central panel) and after 90 days (right panel), highlighting the pronounced desiccation in samples without water addition. Some elements in this figure were created in BioRender. Pena, M. (2026) https://BioRender.com/mmvphi1.

## Differences in abiotic factors modulate microbial species selection

Metagenome sequencing was performed using three biological replicates per treatment (L, LW, P, PW, C, and CW) after 90 days of incubation. This sample selection was supported by detected drastic changes in microbial community structure and diversity after 90 days of incubation (Fig. 2). On average, 65 million reads per sample (ranging from 50 to 150 bp) were obtained, yielding approximately 23.4 Gbp of data per treatment (Supplementary Data 2). Taxonomic classification of metagenomic reads using Kraken2 and Bracken assigned ~27% of high-quality reads to known taxa (Fig. 4AB and Supplementary Data 2). Among the classified reads, 91% corresponded to prokaryotic sequences, with an average of 710 prokaryotic families detected per sample. Bar plot analysis revealed that *Streptomycetaceae* and *Micromonosporaceae* were the most abundant families (~20% of assigned reads) in the L treatment, whereas the halophilic archaeal family *Haloarculaceae* dominated (~15% of assigned reads) in the C treatment. The relative abundance of *Haloarculaceae* was also higher in the P treatment compared with PW and CW treatments (Fig. 4B), indicating that the absence of seawater and soil desiccation favors halophilic archaea, while lignocellulose addition promotes actinobacterial growth. Shannon index analysis of metagenomic read-based taxonomic profiles showed significantly lower diversity in the L treatment and higher diversity in the LW treatment ($p < 0.05$, Fig. 4C). Clustering analysis further revealed distinct microbial community structures among the L, LW, and C treatments (Fig. 4D).

## Microbial diversity captured by metagenome-assembled genomes

High-quality metagenomic reads from each sample were pooled by treatment and subjected to co-assembly, yielding an average of 2.6 million contigs per treatment (N50 ≈ 952 bp). Approximately 28% of high-quality reads mapped to contigs (Supplementary Data 2). The binning process grouped 141,000 contigs into 196 bins (Fig. 4A). After refinement and dereplication, 66 medium-quality MAGs (≥50% completeness, ≤10% contamination) and 8 high-quality MAGs (≥90% completeness, ≤5% contamination) were retained. In total, ~5% of metagenomic reads mapped back to these 74 MAGs, indicating limited diversity capture. The average MAG size was 3.33 ± 1.34 Mbp, with ~535 contigs per genome (Supplementary Data 3). Taxonomic classification with GTDB-Tk assigned 4 MAGs to *Archaea* (P1_*Haladaptatus*, P10_*Halomarina*, CW2_*Nitrososphaeraceae*, and CW3_*Nitrososphaeraceae*), while the remainder were assigned to *Bacteria*. Interestingly, 5 MAGs enriched in the CW and LW treatments were affiliated with the taxa UBA6522 (an unclassified *Gammaproteobacteria*), and 4 MAGs enriched in the P and C treatments were affiliated with UBA5704 (an unclassified *Acidimicrobiia*). Only 18 MAGs were assigned to known and named genera or species in GTDB (Supplementary Data 3).

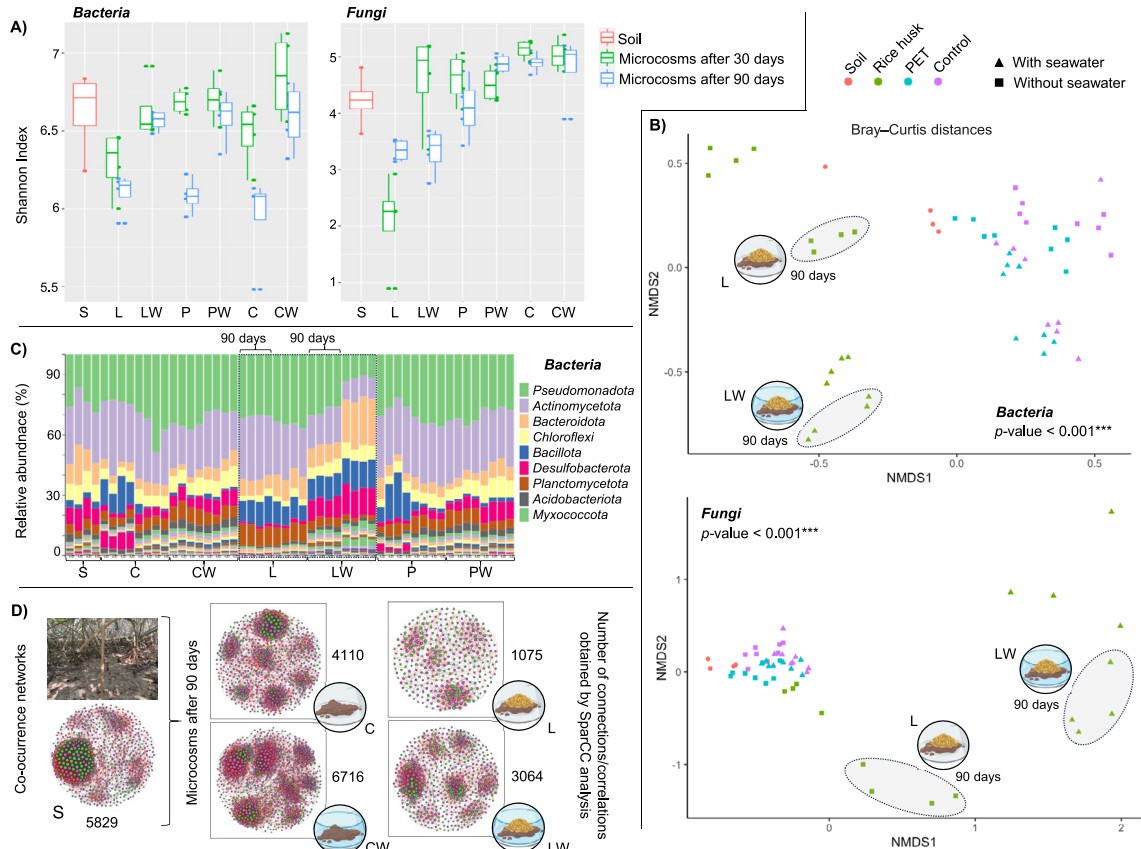

**Fig. 2 | Mangrove soil microbial community restructuring assessed via bacterial 16S rRNA gene and fungal ITS2 amplicon sequencing. A** Box plots showing Shannon diversity index values for bacteria (left) and fungi (right) in the initial soil sample (S, red) and in soil microcosms incubated for 30 days (green) and 90 days (blue). Four biological replicates ($n = 4$) were used per treatment (L, LW, P, and PW), negative controls (C and CW), and in situ soil (S). In box plots, the center bar is the median. The top and bottom lines of each box represented the 25th and 75th quantiles. The whiskers represent: Lower whisker = smallest observed value ≥ Q1 − 1.5 × IQR and Upper whisker = largest observed value ≤ Q3 + 1.5 × IQR. Source data are provided as a Source Data file. **B** Non-metric multidimensional scaling (NMDS) plots based on Bray–Curtis distances illustrating differences in bacterial (top) and fungal (bottom) community composition among treatments. Different colors represent treatments, and shapes indicate the presence (triangles) or absence (squares) of seawater addition. Ellipses highlight the distinct clustering of rice husk treatments (L and LW) after 90 days of incubation. The inclusion of seawater and

the lignocellulose input were primary factors that significantly shifted bacteria (PERMANOVA; adonis2; Df = 2; $R^2 = 0.08429$; $F = 3.7627$; $p < 0.001$) and fungal (PERMANOVA; adonis2; Df = 2; $R^2 = 0.08375$; $F = 3.4989$; $p < 0.001$) communities. **C** Stacked bar plots showing the relative abundance (%) of dominant bacterial phyla in the initial soil sample (S) and in microcosms incubated for 30 and 90 days. Treatments L and LW after 90 days are highlighted. **D** Cross-kingdom fungal–bacterial co-occurrence networks inferred from SparCC analysis in the initial soil sample (left) and in microcosms after 90 days. Numbers indicate the total number of connections (edges) in each network. Purple and green nodes represent bacterial and fungal amplicon sequence variants (ASVs), respectively. The SparCC correlations were based on a magnitude of >0.7 (positive correlation) or <−0.7 (negative correlation), and is statistically significant ($p < 0.01$). The size of the nodes is proportional to the number of connections (degree). Additional raw data is provided in Supplementary Data 1. Some elements in this figure were created in BioRender. Pena, M. (2026) https://BioRender.com/mmvphi1.

## Clustering of genomes provides insight into their provenance and selection in microcosms

Based on read mapping and relative abundance data, the 74 MAGs were clustered into six significantly different groups or functional guilds (Kruskal–Wallis with Bonferroni correction; $p < 0.05$) (Fig. 5AB). Group A included MAGs enriched in the lignocellulose-amended treatments (L and LW). Three MAGs (L1_*Tuberibacillus calidus*, L3_*Bacillus smithii*, and L2_*Cohnella laeviribosi*) were particularly abundant in both treatments. Group B comprised MAGs enriched in CW and PW compared with L and LW, including four affiliated with the taxon UBA6522. Group C contained MAGs that were specifically enriched in L, such as L8_*Streptomyces* and L11_*Myceligenerans*, which displayed markedly higher relative abundances than others in this cluster. Group D represented MAGs more abundant in LW than in L, although without significant differences from other treatments. Groups E and F showed less consistent patterns. Group E contained MAGs with significant differences between treatments but no clear overall trend. Several members of this group, however, were abundant in C, P, and L,

including halophilic taxa such as P10_*Halomarina*, C11_*Actinopolyspora*, P1_*Haladaptatus*, and C12_YR4-1 (an unclassified *Balneolaceae*). Group F comprised MAGs with no significant differences across treatments. Additionally, two MAGs assigned to taxon 70-9 (an unclassified *Actinomycetota*) were highly enriched in one replicate of PW (Fig. 5A).

Functional annotation using KEGG revealed that many MAGs shared broadly similar profiles at the KO identifier level, likely reflecting the large proportion of genes involved in conserved cellular processes (Fig. 5B). Nonetheless, clustering based on Jaccard distances explained 12.01% of the variance and showed significant differences, with MAGs in groups A and B displaying distinct metabolic signatures (PERMANOVA; $p < 0.05$, Fig. 5B). Taxonomic clustering further supported these patterns. For instance, halophilic archaea (P1, P10) and ammonia-oxidizing archaea (CW3, CW2) were positioned adjacent, while *Actinobacteriota* (e.g., order UBA5794) and unclassified *Gammaproteobacteria* (e.g., order UBA6522) were grouped as outliers (Fig. 5B).

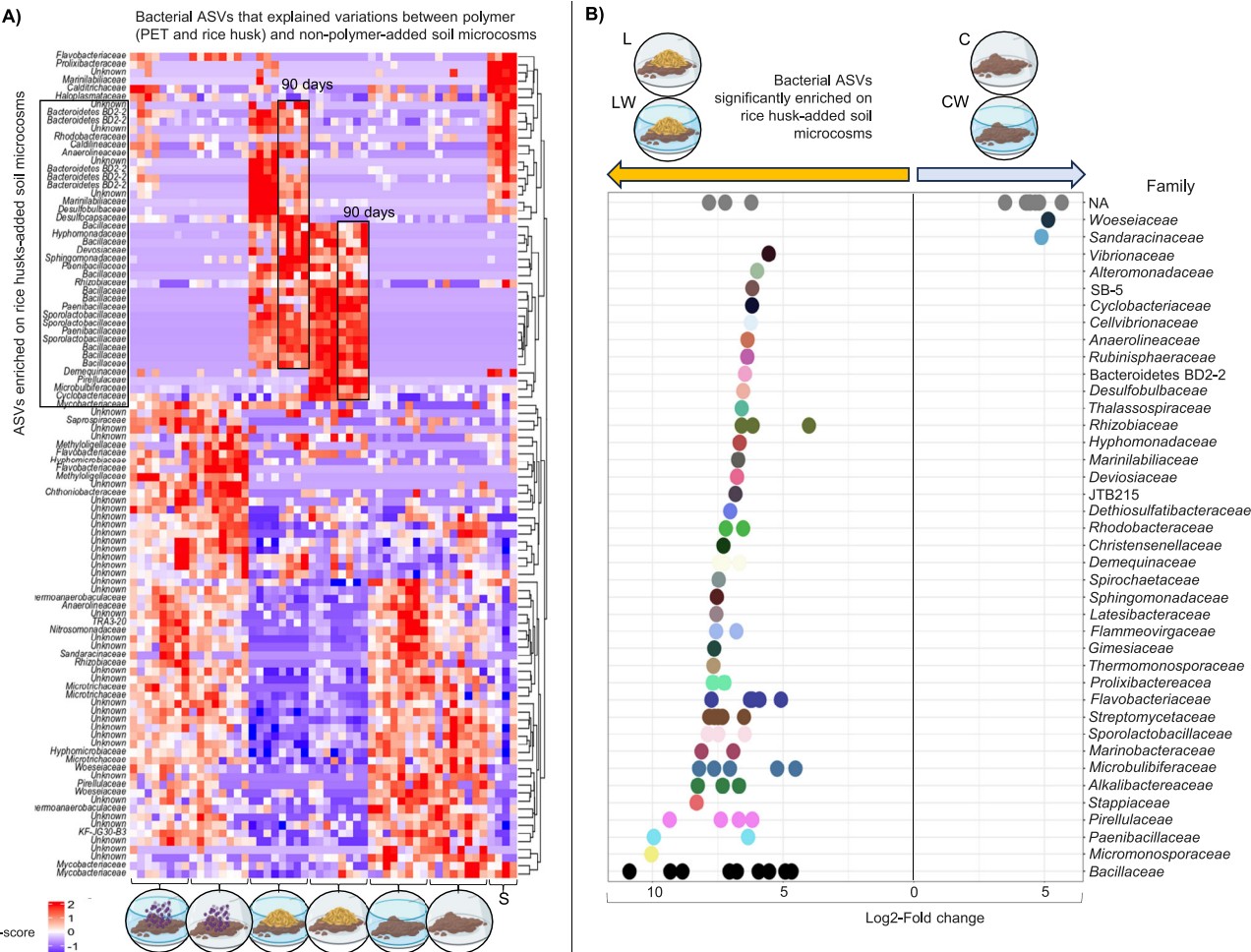

**Fig. 3 | Differential enriched bacterial taxa in lignocellulose-added microcosms. A** Heatmap of bacterial amplicon sequence variants (ASVs) that explained variations between polymer-added (PET and rice husk) from non-polymer-added soil microcosms. These ASVs were identified via the Boruta algorithm. Colors indicate standardized abundance (Z-score), with red representing enrichment and green underrepresentation. ASVs enriched in rice husk-added microcosms (L and LW) after 90 days are highlighted. Source data are provided as a Source Data file.

**B** Volcano plot showing bacterial ASVs significantly enriched (identified via DESeq analysis) in rice husk-added microcosms compared with negative controls. Each point represents an ASV colored by family. The x-axis shows log2-fold change, with higher values indicating stronger enrichment. Icons above illustrate the treatment comparisons, emphasizing rice husk treatments versus control microcosms. Source data are provided as a Source Data file. Some elements in this figure were created in BioRender. Pena, M. (2026) https://BioRender.com/mmvphi1.

## Selection of lignocellulose-transforming microbial species

On average, 2266 (±875) genes were identified per MAG. Functional annotation using eggNOG-mapper revealed that the most abundant COG category was unknown function (Supplementary Data 4). In parallel, proteins from each MAG were annotated against the CAZy database (Supplementary Data 5), which resulted in 3230 genes assigned to 190 families across the 74 MAGs. Analysis of CAZy families involved in lignocellulose degradation (Supplementary Data 5 and Fig. 5C) identified 886 genes within the 74 MAGs, with most overrepresented in groups A and C (Fig. 5A, C). Particularly, L8_*Streptomyces*, L11_*Myceligenerans*, and L14_*Gimesia* carried diverse repertoires of cellulases (e.g., AA10, GH16, GH5, GH6, and GH48), xylanases (e.g., GH10, GH11, GH3, GH43, and GH51), and esterases (e.g., CE1, CE4, CE7, and CE14) (Supplementary Data 5). Three cellulose-degrading families (GH6, CBM4, and CBM16) were significantly enriched in L8_*Streptomyces* (ORA with Bonferroni correction; $p < 0.05$), while four families involved in cellulose and xylan deconstruction (AA10, GH11, GH42, and GH48) were significantly enriched in L11_*Myceligenerans* ($p < 0.05$). Additionally, several MAGs in group A, such as LW3_*Balneolaceae* and LW13_*Latescibacterales*, contained a notable proportion of glycosyl hydrolases targeting plant polymers (e.g., GH10, GH43, GH51, GH67, and GH53) (Supplementary Data 5).

Beyond polysaccharide degradation, we evaluated the presence and enrichment of 60 KEGG orthologs (KOs) linked to the catabolism of lignin-derived compounds. Within group A, genes encoding glutathione peroxidase (K00432), superoxide dismutase (K04564), and catalase (K03781) were significantly enriched in LW12_*Alkalibacter*, L2_*Cohnella laeviribosi*, and L1_*Tuberibacillus calidus*, respectively (ORA-Bonferroni correction; $p < 0.05$, Supplementary Data 6). The capacity to metabolize a range of lignin-derived aromatic monomers was observed across multiple MAGs (Supplementary Data 6 and Fig. 5C). For example, genes encoding enzymes for catechol catabolism were enriched in C11 and LW11 (*catA*), PW1 (*catC*), and C3 and C16 (*dmpB*). All of these MAGs were affiliated with the *Actinobacteriota* phylum (ORA-Bonferroni correction; $p < 0.05$, Supplementary Data 6).

## Screening of putative PET-depolymerizing enzymes within genomes

A total of 26 proteins derived from the 74 MAGs showed matches to enzymes in the PAZy database (>50% amino acid similarity) (Supplementary Data 7). Particularly, two proteins from L8_*Streptomyces* and L11_*Myceligenerans* each had 26 hits against PAZy-derived proteins, displaying high similarity to PET40 (64% similarity, *E*-value 4.44E − 101)

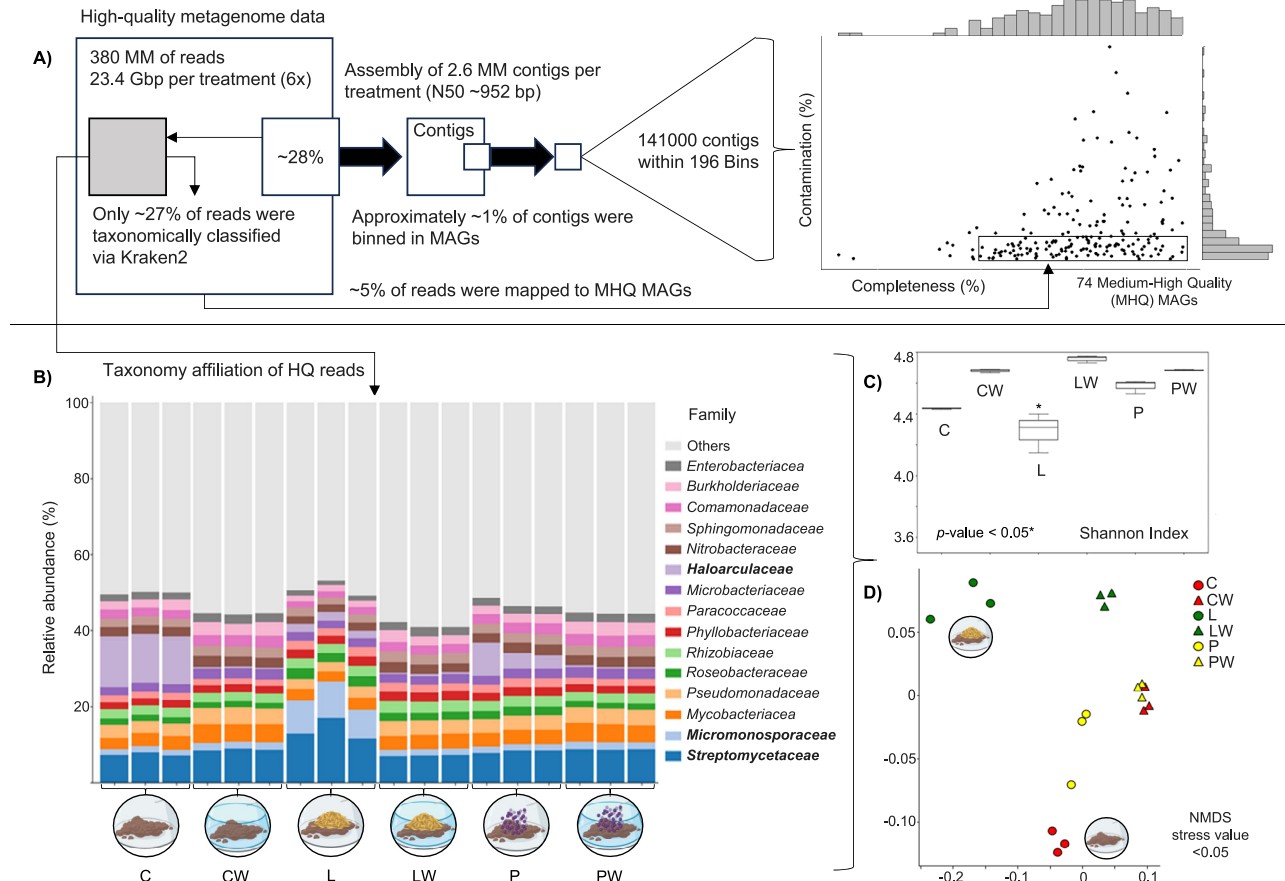

**Fig. 4 | Read and genome-resolved metagenomic analysis in microcosms.**
**A** Schematic overview of metagenomic data processing and assembly. A total of 380 million reads (~23.4 Gbp per treatment) were obtained, of which ~27% were taxonomically classified with Kraken2. About 28% of reads were assembled into contigs, and ~1% of contigs were binned into metagenome-assembled genomes (MAGs). Approximately 5% of reads mapped to medium- to high-quality (MHQ) MAGs. The right panel shows completeness and contamination of the 196 bins obtained, highlighting the 74 MHQ MAGs selected for further analysis. **B** Taxonomic affiliation of high-quality (HQ) reads classified with Kraken2, shown as stacked bar plots representing the relative abundance (%) of bacterial families. **C** Shannon diversity index of HQ raw reads from shotgun metagenome sequencing across treatments (C, CW, L, LW, P, and PW), represented as box plots. Each microcosm treatment included three independent biological replicates (*n* = 3). Box plots show the median (center line) and

interquartile range (IQR; box limits); points represent individual replicates. Mean ± SD values were: C (4.437 ± 0.006), CW (4.682 ± 0.012), L (4.288 ± 0.128), LW (4.759 ± 0.024), P (4.581 ± 0.043), and PW (4.685 ± 0.005). Differences among treatments were assessed using a two-sided Kruskal–Wallis test ($H$ = 16.11, df = 5, $p$ = 0.0065), followed by Dunn's test with Bonferroni correction. A significant difference was detected between L and LW (adjusted $p$ = 0.0086); all other comparisons were not significant. Source data are provided as a Source Data file. **D** Non-metric multidimensional scaling (NMDS) plot (stress < 0.05) based on Bray–Curtis distances, showing differences in community composition among treatments. Points are colored by polymer inclusion and shaped by seawater presence (triangles = with seawater, circles = without). Icons highlight L and C treatments with lower diversity. Source data are provided as a Source Data file. Some elements in this figure were created in BioRender. Pena, M. (2026) https://BioRender.com/mmvphi1.

and Cut190 (69% similarity, *E*-value 1.17E − 118), respectively. In addition, LW4_*Weizmannia coagulans* contained four putative esterases with high similarity to HG-4 (99%), Est8-89 (58%), Est18-23 (61%), and Ces39-5 (60%), with *E*-values ranging from 2.91E − 108 to 4.20E − 229.

**Lignocellulose amendment acts as a resource for putative PET-depolymerizing enzymes**
To broaden the assessment of enzymatic potential beyond the assembled MAGs, we constructed a complete gene catalog—including genes from both binned and unbinned contigs—across all microcosm treatments. In total, 4,250,384 unique proteins were detected. This catalog was screened against the PAZy database using stringent thresholds (E-value ≤ 1E − 15, ≥50% identity, and ≥80% coverage), resulting in 20–120 unique protein matches across treatments (Fig. 6A). Particularly, the lignocellulose-amended treatment (L) yielded the highest number of unique PAZymes, most of which were associated with PLA and PET transformation. In this treatment, approximately 60 putative PETases were detected, primarily affiliated with the *Pseudomonadota*, *Actinomycetota*, *Bacillota*, and

*Halobacteriota* (*Archaea*) phyla (Fig. 6A). Interestingly, PETases from *Halobacteriota* were more frequently detected in dry treatments (P, C, and L). Within the entire gene catalog, only twelve unique proteins met ultra-stringent criteria (*E*-value ≤ 1E − 20, ≥70% identity, and ≥90% coverage). Strikingly, all twelve were found in the L treatment. Among them, five were located in binned contigs affiliated with *Micromonospora* and *Isopthericola*, while four were affiliated with *Marinobacter* and *Microbulbifer* species (Fig. 6B).

**Prediction of properties of putative PET-depolymerizing enzymes**
A sequence-based comparative analysis was performed using two putative PETases identified in MAGs (L8_*Streptomyces* and L11_*Myceligenerans*), eleven putative PETases identified from the complete gene catalog (restricted to *Pseudomonadota* and *Actinomycetota*), and thirteen well-characterized PET-active enzymes (Supplementary Data 8). The SSN indicated that all enzymes clustered together, reflecting high alignment scores, although GlacPETase and HaloPETase1 were the most divergent (Fig. 6C). In the phylogenetic tree, these two enzymes grouped with Mic1 and Mic2 (*Microbulbifer*-derived putative PETases)

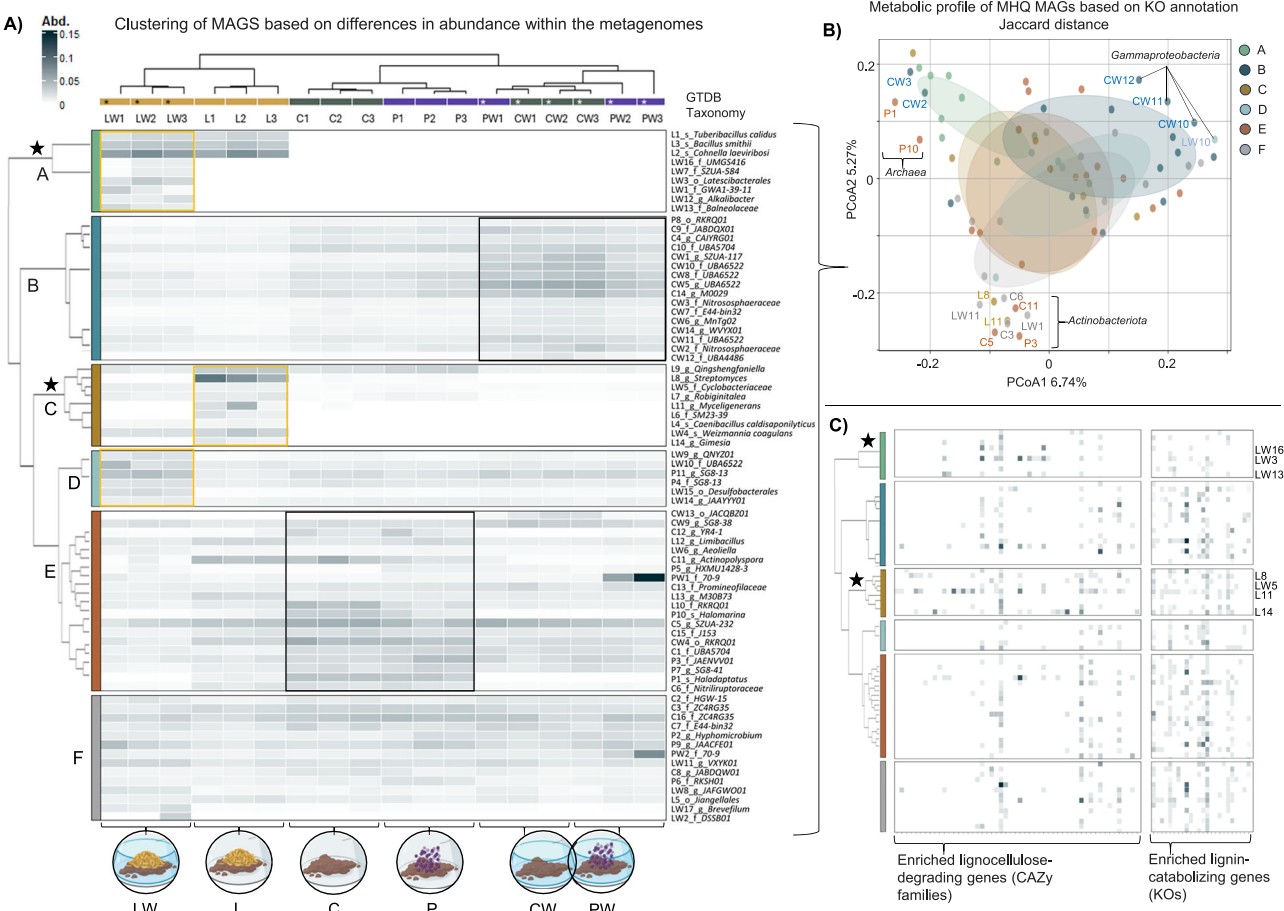

**Fig. 5 | Relative abundance of metagenome-assembled genomes (MAGs) and functional annotation. A** Heatmap showing hierarchical clustering of medium- and high-quality (MHQ) MAGs across treatments, with darker shades indicating higher abundance. MAG taxonomy, assigned using GTDB-tk, is shown on the right. Based on read mapping and relative abundance, the 74 MAGs were grouped into six significant clusters (Kruskal-Wallis test with Bonferroni correction; *p* < 0.05). Icons below indicate treatment type. Source data are provided as a Source Data file. **B** Principal coordinates analysis (PCoA) plot based on Jaccard distances of KEGG ortholog (KO) annotations, illustrating metabolic profiles of MHQ MAGs across the six functional guilds (**A**–**F**). Each point represents a MAG, colored by cluster, and ellipses show cluster distributions. Selected MAGs are labeled to highlight representative taxonomic groups, including *Gammaproteobacteria*, *Actinobacteriota*,

and *Archaea*. **C** Distribution of lignocellulose-catabolizing genes across MHQ MAGs. Heatmaps show enriched lignocellulose-degrading gene families (CAZy, left) and lignin-catabolizing genes (KEGG orthologs, right). While certain MAGs (labeled on the right) are enriched in these genes, the heatmaps depict the distribution across all MAGs grouped by the six clusters. Stars indicate clusters of particular interest. Enrichment was assessed using over-representation analysis (ORA) based on hypergeometric tests applied to CDS-derived functional annotations, with *p*-values adjusted for multiple comparisons using Bonferroni correction; adjusted *p* < 0.05 was considered significant. Source data are provided in Supplementary Data 5 and 6. Some elements in this figure were created in BioRender. Pena, M. (2026) https://BioRender.com/mmvphi1.

(Fig. 6D). Interestingly, Mar1 and Mar2 proteins (from *Marinobacter*) clustered with HaloPETase5 and *H. bauzanensis* PETase, sharing ~72% amino acid similarity. The phylogenetic placement of Mic1/2 and Mar1/2 in the sequence landscape of PET-active enzymes from PAZy is shown in Supplementary Fig. 1. Among the 13 putative PETases, five lacked the functional M5 motif, including two from *Micromonospora* and two from *Microbulbifer* (Supplementary Data 9 and Fig. 6D).

Clustering of predicted 3D structures showed that all type I terrestrial actinobacterial PETases grouped together, except two *Micromonospora*-derived enzymes (Fig. 7A). Additionally, both putative *Microbulbifer*-derived enzymes (Mic1 and Mic2) and HaloPETase1 displayed distinct structural features compared to *Pseudomonadota* homologs (Fig. 7A). Moreover, AI-based prediction of melting temperature (*Tm*) values via ProtScout revealed that three actinobacterial proteins had Tm values between 60 and 67 °C (Fig. 6E; Supplementary Data 10). Furthermore, ProtScout suggested that most putative PETases have an optimal activity at mesophilic conditions, >4% salinity, and near-neutral pH[5–9], except Mar2 from *Marinobacter*, which was predicted to function optimally under alkaline pH[9–14] (Supplementary Data 10 and

Fig. 7B). Multiple sequence and structure alignments confirmed the presence of catalytic triads (SDH), active sites, and disulfide bonds in all proteins (Supplementary Data 11 and Fig. 7B). For example, the catalytic triad of the L8_*Streptomyces* PETase was located at S138, D184, and H216, with an oxyanion hole at Y10 and M139, and a single disulfide bond between C249 and C265 (Fig. 6F). In contrast, both *Marinobacter*-derived proteins and PETase 403 (*Ketobacter* sp.) each contained three disulfide bonds (Supplementary Data 11 and Fig. 7B).

Therefore, ESMfold-predicted 3D structures of Mic1 and Mic2 (Fig. 8A) as well as Mar1 and Mar2 (Fig. 8B) were aligned with crystal structures of their closest homologs: HaloPETase1 (PDB: 9hl5) for Mic1 and Mic2, and IsPETase (PDB: 5xjh) for Mar1 and Mar2 (see Fig. 7 for identification of related structural homologs). Overall, the active site architectures were qualitatively similar across all structures. However, root-mean-square deviation (RMSD) values of aligned backbone Cα atoms indicated greater structural divergence for Mic1 (1.50 Å) and Mic2 (1.23 Å) relative to HaloPETase1, compared with Mar1 (0.68 Å) and Mar2 (0.71 Å) relative to IsPETase (Fig. 8C, D). These global structural differences were also reflected locally at the predicted subsite I

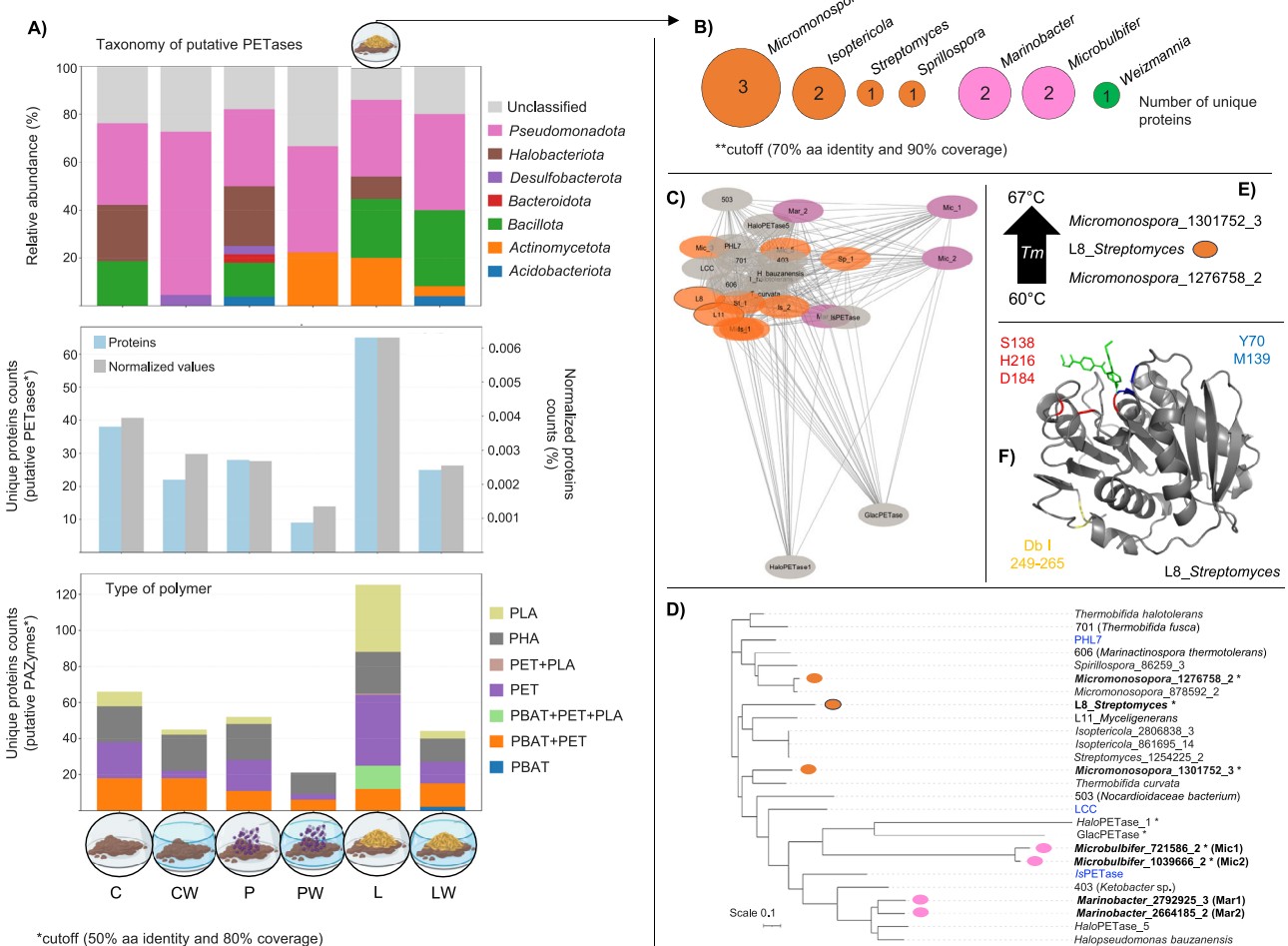

**Fig. 6 | Screening and analysis of PAZymes in metagenomes derived from microcosms. A** Bottom panel: distribution of unique putative PAZy proteins classified by targeted polymer type (polylactic acid PLA, polyethylene terephthalate -PET, and polybutylene adipate terephthalate -PBAT) across treatments (C, CW, L, LW, P, and PW). Middle panel: number of unique putative PETases per treatment (blue bars, left axis) and normalized protein counts (gray bars, right axis). Top panel: relative abundance of phylum-level taxonomic affiliations of putative PETases in each treatment. Source data are provided as a Source Data file. **B** The circles represent the number of unique putative PETases identified at ≥70% identity and ≥90% coverage across bacterial genera in the L treatment. Circle size reflects the number of unique PETases per genus and is colored by phylum (orange: *Actinomycetota*; pink: *Pseudomonadota*; green: *Bacillota*). **C** Sequence similarity

network (SSN) comparing eleven putative PETases identified in the gene catalog from the treatment L (colored by phylum), PETases from MAGs L8 and L11 (outlined in black), and thirteen prototypical PET-active reference enzymes (gray). **D** Phylogenetic tree of the same set of putative PETases and reference enzymes; benchmark enzymes are shown in blue. Asterisks indicate enzymes lacking the functional M5 motif. **E** Predicted melting temperatures (Tm) of three actinobacterial putative PETases, ranging from 60 to 67 °C. **F** Structural model of the L8 *Streptomyces*-derived putative PETase showing key active site residues (S138, H216, D184 in red), oxyanion hole residues (Y70, M139 in blue), and disulfide bonds (Db I, C249–C265 in yellow). Some elements in this figure were created in BioRender. Pena, M. (2026) https://BioRender.com/mmvphi1.

(classification according to [27]). While IsPETase homologs Mar1 and Mar2 showed highly similar residue occupations and spatial positioning of corresponding sidechains (Fig. 8D), HaloPETase1 homologs Mic1 and Mic2 displayed greater variation. Specifically, instead of T88, Q157, and Y180 in HaloPETase1, Mic1 and Mic2 contained Y, M, and a non-canonical H at the equivalent positions. At position L184, both candidates showed a conserved L in a spatially similar orientation. Notably, the positioning of sidechains H and L (corresponding to Y180 and L184 in HaloPETase1) in Mic1 and Mic2 may have been influenced by prediction uncertainty, as indicated by pLDDT scores below 80 (Fig. 8A). However, these lower pLDDT scores may reflect intrinsic conformational flexibility rather than prediction error alone.

## Discussion

Within soil, competition for carbon resources is one of the major mechanisms modulating microbial community assembly[42,43]. In mangroves, soil organic carbon (OC) supports microbial growth and

originates from litterfall and the rhizosphere or is transported by tides and rivers from marine or terrestrial sources (e.g., agricultural runoff)[44]. Plant biomass contains a mixture of labile carbon sources and recalcitrant polymers that are dynamically degraded over time[45]. In our microcosm experiment, the rice husk substrate was partially consumed after 90 days of incubation (Fig. 1B), and the remaining plant biomass likely represent the hardest fraction to be degraded (i.e., lignin). The input of lignocellulose into mangrove soils can alter fungal-bacterial dynamics and species interactions[46,47], intensifying ecological competition among taxa and often leading to reduced diversity, consistent with our observations in the L treatment (Figs. 2AD and 4B). This decline in microbial diversity may be linked to the presence of plant-derived oligo- and monosaccharides, which favor the enrichment of specific populations. However, once these external resources are depleted, soil microbial communities are expected to recover and follow successional trajectories similar to those occurring in natural environments[48,49].

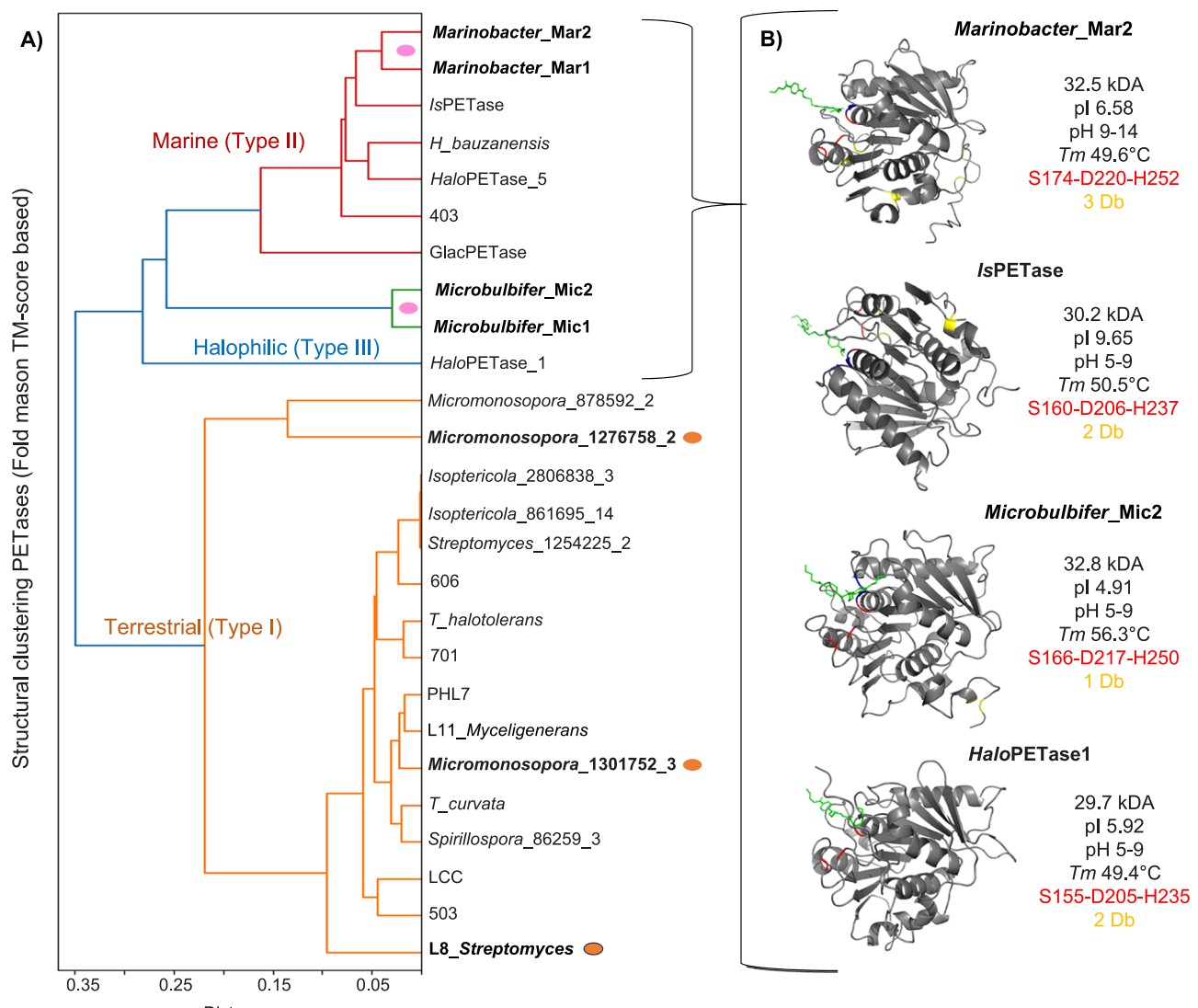

**Fig. 7 | Structural analysis of putative PETases identified in lignocellulose-added microcosm (L treatment). A** Clustering of putative PETases and reference enzymes based on 3D structural similarity (FoldMason TM-score), colored by PETase type group. Notable PETases are highlighted in bold. **B** Predicted 3D structures of representative PETases: *Marinobacter* 2664185-2 (Mar2), IsPETase, *Microbulbifer* 1039666-2 (Mic2), and HaloPETase1. Annotated molecular weight, isoelectric point (pI), pH range, and predicted melting temperature (Tm) are shown. Active site residues are highlighted in red and disulfide bonds (Db) in yellow.

Following lignocellulose input, seawater addition was identified as a second factor restructuring mangrove soil microbial communities within the microcosms. Although water content was not quantified, desiccation was evident in the flasks by the end of the experiment (Fig. 1). The gradual evaporation of seawater and the associated increase in salinity acted as strong selective pressures, particularly in the L treatment (Figs. 2A and 4C). Salinity exerts a relatively uniform selective pressure on soil microbial taxa, often outweighing the influence of other environmental variables such as temperature and pH[50,51]. Moreover, seawater addition appeared to buffer diversity loss, especially after 90 days of incubation. In general, microcosms receiving seawater amendments displayed higher diversity than those without seawater, likely due to the introduction of distinct marine-derived taxa[52]. Seawater may also decrease the concentration of salt and reduce oxygen diffusion within the soil microcosms, potentially constraining aerobic decomposition of OC[53,54]. Overall, our findings support that flooding events, carbon input, drought, and salinity are major drivers of structural shifts in mangrove soil microbial communities, similar to patterns observed in other terrestrial systems[9,55,56].

Although static soil microcosms do not fully replicate the dynamic conditions found in nature, they provide a valuable framework for investigating how microbial communities respond to multiple environmental disturbances (e.g., drought, salinity, and external carbon inputs). They also serve as a practical strategy for selecting functional target microbial populations. As mentioned, the incubation conditions favored the enrichment of spore-forming, halophilic, and lignocellulolytic taxa such as *Bacillaceae*, *Micromonosporaceae*, and *Streptomycetaceae* in the L treatment (Fig. 3AB). These groups are well known for their metabolic versatility, tolerance to high salt concentrations, and ability to thrive in OC-rich environments such as mangrove soils[57–59]. Dormancy and spore formation are widely documented adaptive microbial strategies to cope with abiotic stresses such as salinity and drought. For instance, in salt marsh sediments, dormant taxa can comprise more than half of the microbial community[60]. Although salinity can compromise OC processing and inhibit hydrolytic enzyme activity[61], certain bacterial taxa (e.g., *Planctomycetaceae*, *Balneolaceae*, *Microbulibiferaceae*, and *Alkalibacteraceae*) were able to tolerate both desiccation and salinity while

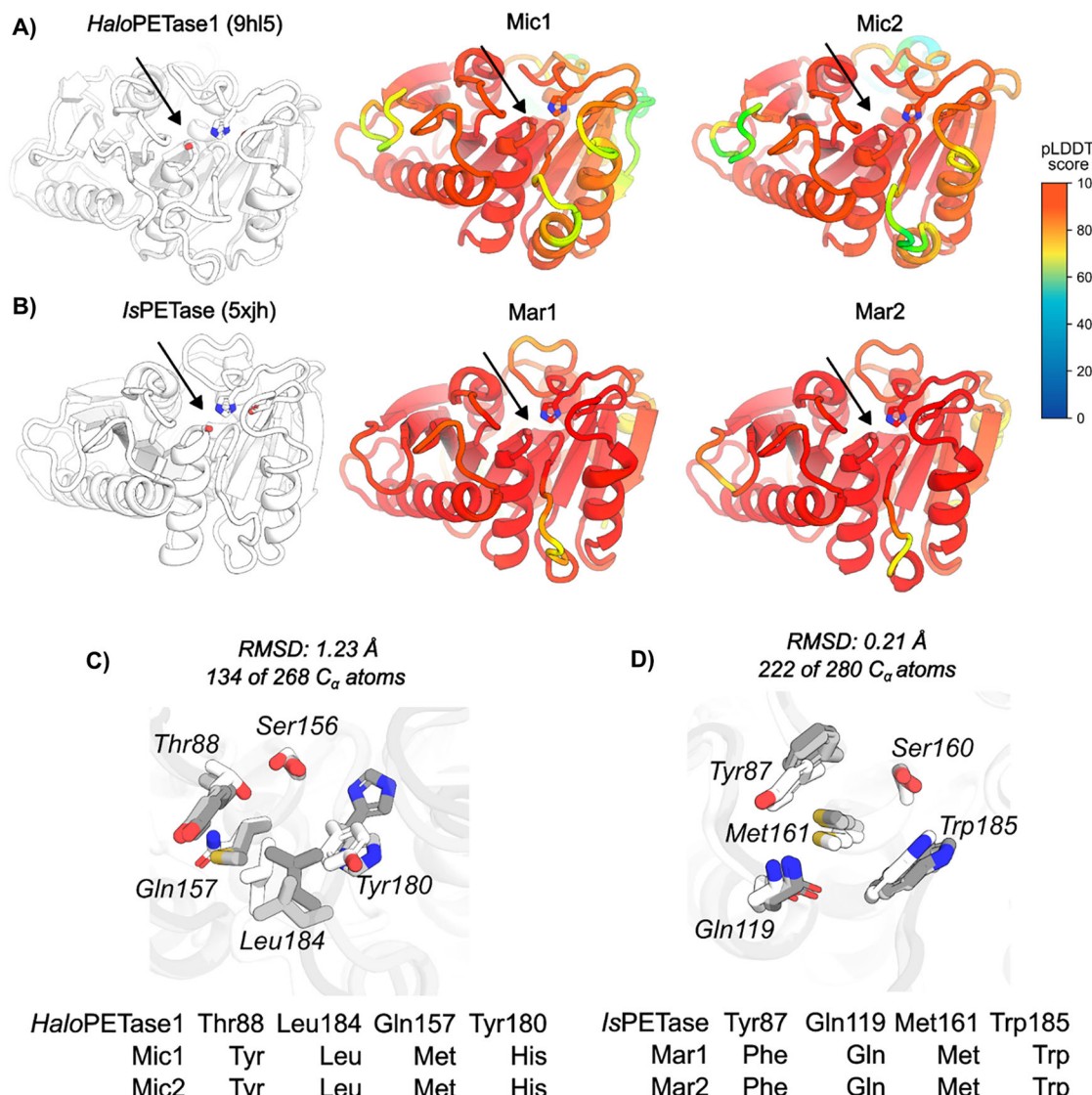

**Fig. 8 | Structural comparison of putative PETases (Mar1, Mar2, Mic1, and Mic2) with HaloPETase1 and IsPETase at subsite I. A, B** Crystal structures of HaloPE-Tase1 (PDB: 9hl5; light gray) and IsPETase (PDB: 5xjh; light gray) overlaid with ESMfold-predicted structures of Mar1, Mar2, Mic1, and Mic2, color-coded by predicted local distance difference test (pLDDT) scores. Active sites are indicated by black arrows, showing the canonical Ser-His-Asp catalytic triad (stick representation). **C, D** Subsite I amino acid occupation (see also Joo et al.[27] for PETase classification) for HaloPETase1 (HP1) and homologs Mic1 and Mic2 in (**C**) and for IsPETase and homologs Mar1 and Mar2 in (**D**) (see Fig. 7 for identification of closest structurally related PETases). Backbone Cα atoms were superimposed in PyMOL using the "align" function with outlier rejection of 2 Å over 5 cycles.

retaining the genetic potential to metabolize plant-derived carbon. Members of *Microbulibiferaceae* and *Planctomycetaceae*, in particular, are well-recognized contributors to the degradation of complex polysaccharides in marine systems[62–64]. Additionally, chemoorganotrophic and halophilic *Archaea* affiliated with the order *Halobacteriales* were enriched across all microcosm treatments without seawater, particularly in C and P compared with L (Fig. 4B). This supports previous findings showing that these archaeal taxa respond positively to increased desiccation and salinity[65,66]. Collectively, these patterns indicate that specific bacterial and archaeal taxa were selectively enriched in the microcosms due to their capacity to withstand the imposed environmental constraints. We acknowledge that higher-resolution insights into their temporal dynamics and functional activity would require increased sampling frequency coupled with metatranscriptomic analyses.

Metagenomes were analyzed using three complementary strategies—based on reads, genes, and genomes—each providing distinct insights into the data[67]. Although assembly-based metagenomic analyses provide only a partial view of the total diversity in soil, they enable the development of gene and genome catalogs that are essential for characterizing the functional potential of microbiomes[55,68]. Furthermore, the reconstruction of MAGs yields valuable information on abundant species, improving taxonomic resolution and enabling a more precise link between taxa and potential metabolic processes. In this study, most MAGs recovered from the microcosms were affiliated with unnamed or unclassified prokaryotic taxa (e.g., UBA6522, UBA5794, HK1, 70-9, and JACQBZ01) (Supplementary Data 3). This finding is consistent with soil-derived microbiomes, where many prokaryotic species remain undescribed[69]. Taken together, the relatively low proportion of taxonomically annotated reads (~27%), assembled reads (~28%), and reads mapped back to MAGs (~5%), along with the large fraction of genes assigned to unknown function, highlight the vast and still underexplored microbial diversity and functional novelty present in mangrove-associated microbial communities. It is important to note that fungi also possess substantial enzymatic potential (e.g., cutinases and esterases) for the degradation

of polyesters. However, this component was not explored in the present study due to the very low relative abundance of fungal-derived sequences (≤0.2%) detected among the unassembled reads in most samples, which limited the robustness of downstream analyses.

In this study, we did not observe a significant enrichment of PETases in the PET-amended treatments (Fig. 6A). This suggests that PET-added microcosms did not favor the selection of PET-degrading taxa, likely due to the recalcitrance of PET to depolymerization, the availability of alternative carbon sources, or harsh environmental conditions. Interestingly, higher numbers of predicted PETases were detected in the L treatment, indicating that the inclusion of lignocellulosic substrates promotes the selection of prokaryotic species with an expanded genetic potential for PET depolymerization. Using lignocellulose as a selective factor could therefore be a powerful strategy to enrich PET-degrading enzymes from environmental microbiomes. Indeed, one of the most important PETases currently used in industrial applications (i.e., LCC) was originally detected from a plant biomass–rich environment (leaf-branch compost)[70,71]. In this present study, ten to twelve putative PETases were detected in the complete gene catalog and not within the set of high-quality MAGs, suggesting that these enzymes are present in microorganisms whose low relative abundance constrains MAG recovery. This finding reinforces the use of both gene-centric and genome-resolved metagenomics as complementary approaches for prokaryotic enzymes bioprospection and discovery.

In microcosms without seawater, we observed a selection of putative PETases from *Halobacteriota*. To date, only one PET-active enzyme (PET46) from Candidatus Bathyarchaeota archaeon has been described[72]. Thus, a detailed characterization of these putative halophilic archaeal PETases will be important to determine whether PET depolymerization is more broadly distributed among archaeal taxa. Furthermore, under a strict protein similarity cutoff, we identified 7 and 4 putative PETases of terrestrial (*Actinomycetota*) and marine (*Pseudomonadota*) origin, respectively, in the L treatment[25,73]. The detection of signal peptides and the presence of the catalytic triad in all putative PETases suggest that they may be secreted and functionally active on PET. Notably, three actinobacterial candidates (one from *Streptomyces* and two from *Micromonospora* species) showed predicted Tm values between 60–67 °C. This is particularly relevant for industrial applications, as PET conversion becomes efficient when temperatures approach its glass transition temperature of ~70 °C[26,32,33].

Halophilic PET-active enzymes have been detected in diverse microorganisms, including *Halopseudomonas*, *Vibrio*, *Streptomyces*, and in metagenomes from deep-sea sediments[28,30,74,75]. However, no active halophilic PETases have been reported from mangrove-associated microbial communities or *Microbulbifer* species. In contrast, *Marinobacter* is already recognized as a PET- and PBAT-degrading microorganism[37,76]. Here, we identified two putative PETases from *Marinobacter* that contain three disulfide bonds, an unusual feature also reported in PETases from *Vibrio* and Antarctic metagenomes[75,77]. This trait has been proposed as a structural adaptation to support enzyme activity at low temperatures and under high-salinity conditions[78,79]. Structural alignments between these putative *Marinobacter* PETases and their homologs revealed catalysis-relevant residues at the active sites. Based on this analysis and phylogenetic inference, the *Marinobacter*-derived PETases (Mar1 and Mar2) appear to represent type II PETases.

Two additional putative PETases from *Microbulbifer* species displayed highly dissimilar sequences and structural conformations (Figs. 7 and 8) compared with known PET-active enzymes. This strongly suggests that these enzymes may represent a new group of PETases, which should be further validated through enzymatic activity assays. Interestingly, these putative *Microbulbifer* PETases did not contain an intact functional M5 motif, which has been predicted to be important for PETase activity[73]. However, this motif is also absent in

HaloPETase1, whose PET activity has been experimentally confirmed[28], as well as in other PET-active enzymes derived from hot springs[80] and glaciers (GlacPETase)[81]. This evidence suggests that the M5 motif may not be essential for PETase activity and could be absent in newly proposed families of PET-active enzymes. Finally, structural comparisons revealed that Mic1 and Mic2 differed from HaloPETase1 at subsite I. This flexibility, particularly around subsite I, has been shown to enhance substrate recruitment and product release, suggesting that the observed structural variability in Mic1 and Mic2 could be functionally relevant for their catalytic activity. Previous studies have highlighted the role of flexible regions surrounding the catalytic site in active PETases[82,83]. These features make Mic1 and Mic2 promising candidates for further characterization, with potential to expand the currently narrow sequence space of PET-active enzymes.

While some of the identified bacterial-derived putative PETases identified in this study shared similarity with known PET-active enzymes, their origin from saline environments suggests they may exhibit halotolerance. Additionally, several actinobacterial-derived putative PETases were predicted to be thermostable, making them promising starting scaffolds for protein engineering aimed at enhancing activity under elevated temperatures. Such traits highlight their potential application in advanced PET biocatalysis strategies, with direct relevance to circular bioeconomy goals and marine bioremediation efforts. The enzymatic activity of these putative PETases should be experimentally validated and benchmarked against relevant reference enzymes (e.g., PES-H1$^{L92F/Q94Y}$, HotPETase, and LCC$^{ICCG}$) to rigorously assess their functional performance and biotechnological potential in industrial scenarios. Collectively, our findings underscore the value of integrating ex situ microbiome perturbation experiments with metagenomics, protein structural analysis, and AI-based predictions as a powerful approach to discover, characterize, and prioritize putative novel PETases from environmental microbiomes.

## Methods

### Soil microcosm experiments

A laboratory-based microcosm experiment was conducted to induce shifts in the structure of mangrove soil microbial communities using inputs of lignocellulose (rice husk) and polyethylene terephthalate (PET) particles under two conditions: with and without seawater (collected near the soil sampling site). Before setting up the microcosms, rice husk was knife-milled through a 1-mm screen. PET particles (<5 mm; Eastar™ Copolyester GN071) and rice husk particles were washed twice with distilled water and 70% (vol/vol) ethanol, then air-dried at room temperature. A total of 25 soil samples (~500 g each) were collected from the top 10 cm of soil at the Barú peninsula (Cartagena de Indias, Colombia; 10°15′48.5″N, 75°35′29.8″W) (Fig. 1). To capture environmental heterogeneity, samples were taken at random intervals along a ~50 m transect extending inland from the shoreline. Soil and seawater sampling was conducted under permit number 2021123746-1-000 issued by the Autoridad Nacional de Licencias Ambientales (ANLA). Fresh mangrove soil samples were homogenized, sieved through a 2-mm mesh, and combined into a single bulk sample. Each experimental unit was prepared in a 100-mL Erlenmeyer flask containing 15 g of homogenized mangrove soil. An external carbon source (1.5 g), either rice husk (L treatment) or PET particles (P treatment), was then added at a final concentration of 10% w/w. Additionally, 15 mL of seawater was added to half of the experimental units, yielding the LW and PW treatments. Control microcosms with seawater (CW) or without seawater (C) received no additional carbon source. All microcosm treatments (L, LW, P, PW, C, and CW) were incubated at 30 °C for 30 and 90 days. Each treatment was replicated four times, resulting in a total of 48 experimental units (*n* = 48) (Fig. 1). After incubation, soils from the microcosms were homogenized and collected for DNA extraction using the DNeasy PowerSoil Kit (Qiagen, Maryland, USA), following the manufacturer's instructions. DNA from

the original (in-situ) homogenized soil was also extracted for comparison.

## Microbial diversity analyses based on 16S rRNA gene and ITS2 amplicon sequencing

DNA extracted from soils was sent to Macrogen, Inc. (Seoul, Korea) for bacterial 16S rRNA gene amplicon sequencing of the V3–V4 region (primers Bakt-341F and Bakt-805R[84]) and fungal internal transcribed spacer (ITS) 2 region (primers ITS3 and ITS4) using the Illumina MiSeq platform (300 bp paired-end reads). Raw sequences were processed with the DADA2 pipeline v1.26[85]. Briefly, primers were removed using Cutadapt[86], after which sequences were filtered and trimmed, errors were identified and corrected, and paired sequences were merged and dereplicated before taxonomy was assigned using the SILVA rRNA gene database (v.138) for bacteria and the UNITE database (v.10.5.2021) for fungi. The processed sequences were imported into PhyloSeq (v1.48.0), and non-fungal and non-bacterial reads were removed. Amplicon sequencing variant (ASV) tables were rarefied to 10,000 (16S rRNA gene) or 5000 (ITS2) reads for downstream analyses, including comparisons of alpha and beta diversity metrics and statistical tests (e.g., PERMANOVA). All alpha diversity tests were conducted using ANOVA followed by Tukey's HSD for pairwise comparisons (alpha = 0.05). ASVs associated with distinct treatment groups were then identified using the Boruta algorithm[87] and DESeq2 v1.5.0.2[88]. In particular, the Boruta algorithm was implemented in R and used to detect the features (i.e., ASVs) that explained variations between the treatment groups (i.e., seawater addition or lignocellulose addition). Briefly, bacterial ASVs indicative of treatment groups were modeled independently using random forest analysis with 1000 trees, followed by feature selection using the Boruta algorithm. The abundance of significant ASVs (e.g., importance > 2) was then $z$-score transformed and visualized with heat maps built using ggplot2, as is reported in ref. 24. Differential abundance analysis was performed parametrically using the Wald test. The statistical significance of differentially abundant taxa was determined at $p = 0.05$ after FDR multiple comparisons corrections.

Additionally, nonrandom co-occurrence analyses were performed using SparCC[89]. To this end, we selected bacterial and fungal ASVs with at least 30 sequences, which together accounted for more than 90% of the total reads. For each co-occurrence network, $p$-values were estimated from 99 permutations of randomly shuffled data tables. Only SparCC correlations with an absolute magnitude >0.7 or <−0.7 and statistical significance ($p < 0.01$) were retained for network construction. In the resulting networks, nodes represent ASVs, and edges denote strong and significant correlations. Network topology was assessed using metrics such as the number of nodes and edges, average degree, clustering coefficient, modularity, and network density. Network visualizations were generated in Gephi v0.1[90].

## Metagenome sequencing and read-based taxonomic classification

Three biological replicates from each treatment (i.e., L, LW, P, PW, C, and CW) incubated for 90 days ($n = 18$) were selected for metagenomic sequencing and analysis (Fig. 1). High-quality microbial DNA was sequenced at Novogene (Cambridge, UK) on the Illumina NovaSeq platform (2 × 150 bp; Illumina Inc., San Diego, CA, USA). The quality of raw reads was first assessed using FastQC v0.11.9[91]. Reads were then quality-filtered and trimmed with Trimmomatic v0.36 (minimum PHRED quality score of 30 and minimum length of 50 bases[92]). Taxonomic profiling was performed on unassembled high-quality reads using Kraken2 v1.4.0[93], with Bracken v2.9[94] used to estimate relative abundances. The PlusPFP database (version 2024-09-04), which includes RefSeq entries for *Archaea, Bacteria, Eukarya*, viruses, and plasmids, served as the reference. Alpha and beta diversity metrics, including the Shannon index and Bray–Curtis dissimilarity, were calculated from the unassembled read-based taxonomic profiles using the vegan package in R[95]. Complementary statistical analyses and visualizations were carried out in Python using the scipy[96] and scikit-bio v0.5.9[97] libraries.

## Reconstruction and analysis of metagenome-assembled genomes

For each treatment, high-quality and clean reads were co-assembled into contigs using the de novo assembler MEGAHIT v1.1.3 with the default $k$-mer range[98]. Assembly quality was evaluated with QUAST v5.0.2[99]. Clean reads were mapped to contigs using Bowtie2 v2.4.5[100], and contigs were binned with MaxBin2 v2.2.7[101], MetaBAT2 v2.15[102], and VAMB v4.1.3[103], using default parameters and excluding contigs shorter than 1000 bases. Bin refinement across treatments was performed using DAS Tool v1.1.4[104], and bin quality was assessed with CheckM v1.2.1[105]. MAGs were then selected by retaining only medium- and high-quality bins (completeness ≥ 50% and contamination ≤ 10%), following the MIMAG standard[106]. MAGs were taxonomically classified using GTDB-Tk v1.4.0[107] using the GTDB R214 release (2023-04-28). Relative abundance of MAGs across samples was estimated with CoverM v0.7.0 (https://github.com/wwood/CoverM) by mapping quality-filtered reads against the entire MAG catalog. Since MAGs were recovered across all treatments rather than being exclusive to specific microcosms, a new grouping was established using co-abundance clustering. Kruskal–Wallis tests were first applied to relative abundance data to identify MAGs showing significant differences among treatments. Bonferroni correction was applied when conducting multiple comparisons. When significant effects were found, Dunn's post hoc test was performed to assess pairwise differences. Finally, hierarchical clustering of MAGs was conducted using Ward's linkage method to minimize intra-cluster variance. The resulting dendrogram defined clusters of co-abundant MAGs, which were subsequently used as categorical variables in downstream MAG-level analyses.

## Functional annotation of genes derived from metagenome-assembled genomes

For each MAG, protein-coding sequences (CDSs) were predicted using the DFAST pipeline v1.2.6[108], with Prodigal[109] employed for gene calling. Functional gene annotation was performed with eggNOG-mapper v2 using both the Clusters of Orthologous Groups (COG) and KEGG databases. To specifically evaluate lignocellulose-degrading potential, genes involved in lignin-transforming metabolic pathways were identified from KEGG based on curated gene sets reported by Díaz-García et al.[110]. The presence of carbohydrate-active enzymes (CAZymes) was assessed using the dbCAN3 meta server[111]. To assess the potential for plastic degradation, predicted CDSs were aligned against the PAZy database[112] using DIAMOND v0.9[113] with a minimum amino acid identity threshold of 50%. To account for genome incompleteness, gene counts from COG, KEGG, CAZy, and PAZy annotations were normalized by dividing each value by the corresponding MAG completeness proportion, following the recommendations of Eisenhofer et al.[114]. Functional divergence among MAGs was evaluated via principal coordinates analysis (PCoA) based on Jaccard distances of KEGG-annotated gene profiles (KO IDs), allowing visualization of functional clustering across MAG categories. PERMANOVA was performed to statistically test the significance of these functional differences. Results were visualized as ordination plots, with each point representing an individual MAG. Additionally, over-representation analysis (ORA) was conducted for genes associated with (hemi)cellulose and lignin transformations. Enrichment significance was determined using hypergeometric distribution tests with Bonferroni correction for multiple comparisons. All analyses and visualizations were carried out in Python, employing the scipy[96] and scikit-bio v0.5.9[97] libraries.

## Generation of a gene catalog from metagenomes and screening of plastic-active enzymes

To generate a comprehensive protein catalog from the microcosm-derived metagenomes, the open-source software VEBA was used[115]. Briefly, raw sequences were co-assembled and binned using default workflows and parameters. Binned and unbinned contigs were taxonomically classified using GTDB-Tk[107] via the VEBA pipeline. Gene prediction was performed with Prodigal[109], and the resulting gene models (GFF3 format) were modified to include gene and contig identifiers for downstream analyses. Genes were identified on both binned and unbinned contigs to ensure no information was lost, resulting in a complete gene catalog from all metagenomes. All predicted proteins were aligned against the PAZy database, which contains enzymes with experimentally validated activity on plastic polymers[112], using DIAMOND v0.9[113]. Two thresholds of amino acid sequence identity were applied: (*i*) high confidence (*E*-value ≤ 1E-15, ≥50% identity, and ≥80% coverage) and (*ii*) ultra-high confidence (*E*-value ≤ 1E-20, ≥70% identity, and ≥90% coverage). Unique protein counts were assigned to specific plastic polymers and normalized based on the total number of proteins in each treatment (L, LW, P, PW, C, and CW). Taxonomic affiliation of predicted plastic-active enzymes (PAZymes) located on binned contigs was determined using GTDB-Tk v1.4.0, while those on unbinned contigs were assigned based on the best NCBI BLAST hit.

## Analysis of predictive PET-depolymerizing enzymes

Predicted PETases from the *Pseudomonadota* and *Actinomycetota* phyla that exhibited ultra-high similarity to enzymes in the PAZy database were analyzed in comparison with a set of known wild-type PET-active enzymes (e.g., PHL7, LCC, and IsPETase), including Halo-PETase 1 and 5[28] (Supplementary File 1), as well as two putative PETases detected in MAGs L8 and L11. Sequence Similarity Networks (SSNs) were generated using The Enzyme Function Initiative–Enzyme Similarity Tool[116] with an E-value threshold of $1E-10$ and a sequence identity cutoff of 35% (nodes representing sequences sharing >35% identity were connected by edges, defining SSN clusters). The resulting SSN was visualized using Cytoscape[117]. Multiple sequence alignment (MSA) was performed with MUSCLE[118], and phylogenetic trees were inferred using FastTree 2[119] and visualized with iTOL[120]. BLASTp was employed to compare sequences against the PETase from *I. sakaiensis* to identify the large functional M5 motif[73]. Signal peptides were predicted using SignalP 6.0[121] and removed prior to 3D structure prediction. For cases with no SignalP prediction, PrediSI[122] was used as an alternative. Three-dimensional structures were modeled via ESMFold, and corresponding sequence embeddings were extracted from the ESM-2 model[123]. LE–His tags and predicted signal peptides were excised before docking. Molecular docking of a PET dimer substrate was performed using DiffDock-L[124] to generate plausible ligand–protein binding poses. Catalytic residues were identified by mapping conserved active-site positions from IsPETase onto the MSAs, guided by published structural data[27]. Enzyme biophysical properties were ranked using the ProtScout Python package (https://github.com/new-atlantis-labs/ProtScout), incorporating AI-based models to predict melting temperature ($Tm$) via TemBERTure[125], optimal temperature, pH, and salinity via GeoPoc[126], and solubility via GATSol[127]. Kinetic parameters ($k_{cat}$ and $K_M$) for PET dimer hydrolysis were predicted using CatPred[128], a machine-learning tool trained on enzyme kinetics datasets. Classical physicochemical properties (e.g., molecular weight and isoelectric point) were computed using Biopython[129]. Pairwise structural similarity was quantified with TM-align[130] to obtain TM-scores and RMSDs, which were then used as a distance metric for hierarchical clustering of the structures.

## Reporting summary

Further information on research design is available in the Nature Portfolio Reporting Summary linked to this article.

## Data availability

Raw sequencing data (16S rRNA gene, ITS2, and metagenomic) and metagenome-assembled genomes are available through the European Nucleotide Archive (ENA) under BioProject ID PRJEB72453. Source data are provided with this paper. Additional data generated in this study are provided in the Supplementary Information/Source Data file. Source data are provided with this paper.

## Code availability

Custom scripts for processing, analysis, and visualization of shotgun metagenomic data are available at https://github.com/mariafpv/LignoMangrove-MAGs and at Zenodo: https://doi.org/10.5281/zenodo.18651101 [131]. Code for the structural and physicochemical characterization of putative PETases is available at https://github.com/Robaina/Mangrove-PETases and at Zenodo: https://doi.org/10.5281/zenodo.18656903 [132].

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

## Acknowledgements

We thank the Faculty of Sciences at the Universidad de los Andes (Colombia) for financial and administrative support. DNA samples for sequencing were exported under ANLA permit number 2784. Computational analyses were partially conducted using the ExaCore—IT Core Facility at the Vice Presidency for Research and Creation, Universidad de los Andes. This study was partially funded by the FAPA project (PR.3.2018.5287) awarded to D.J.J. at the Department of Biological Sciences, Universidad de los Andes, and by baseline resources (BAS/1/1096-01-01) provided by A.S.R. at KAUST. A.V. received funding from the Dutch Research Council (grant VI.Veni.212.029). O.T. acknowledges funding from the Deutsche Forschungsgemeinschaft (DFG, German Research Foundation)—Project Number 391977956–SFB 1357 Microplastics, subproject C03 at Bayreuth University. We thank Josh L. Espinoza for assistance with metagenomic analyses. O.T. thanks Prof. Birte Höcker for her supervision. We are also grateful to Intikhab Alam at KAUST for M5 motif predictions.

## Author contributions

M.F.P.V.: Metagenomic analyses, statistical analyses, manuscript revision, and figure design. S.R.E.: Development of gene catalogs, screening of PET-active enzymes, protein characterization, and writing. G.F.C.: Amplicon sequencing analysis, statistical analyses, and writing. O.T.: Protein analyses and writing. F.S.: Sampling, microcosm setup, and computational analyses. L.W.M.: Co-occurrence analysis and writing. C.R.L.: Sampling, experimental design, and microcosm setup. J.G.: Development of gene catalogs, screening of PET-active enzymes, protein characterization, and writing. A.V.: Funding acquisition and writing/revision of the final draft. F.D.A.: Advising, computational analyses, and writing/revision of the final draft. A.S.R.: Funding acquisition, coordination, and writing/revision of the final draft. A.R.: Advising, funding acquisition, computational analyses, coordination, and writing/revision of the final draft. D.J.J.: Sampling, experimental design, microcosm setup, conceptualization, figure design, funding acquisition, project coordination, and writing of the first and final drafts of the manuscript.

## Competing interests

The authors declare no competing interests.

## Additional information

[1]Department of Biological Sciences, Universidad de los Andes, Bogotá, Colombia. [2]NewAtlantis Labs, Inc., Wilmington, DE, USA. [3]Department of Natural Sciences, The University of Maryland Eastern Shore, Princess Anne, MD, USA. [4]Department of Biochemistry, University of Bayreuth, Bayreuth, Germany. [5]Cell and Molecular Biology Laboratory, Center for Nuclear Energy in Agriculture, University of São Paulo, Piracicaba, Brazil. [6]Chemical and Biological Studies Group, Basic Sciences Faculty, Universidad Tecnológica de Bolívar, Cartagena de Indias, Colombia. [7]Department of Marine Microbiology and Biogeochemistry, Royal Netherlands Institute for Sea Research (NIOZ), Hoorn, The Netherlands. [8]Department of Plant Science and Huck Institutes of the Life Sciences, The Pennsylvania State University, University Park, PA, USA. [9]One Health Microbiome Center, The Pennsylvania State University, University Park, PA, USA. [10]Biological and Environmental Sciences and Engineering Division (BESE), King Abdullah University of Science and Technology (KAUST), Thuwal, Kingdom of Saudi Arabia. [11]These authors contributed equally: María Fernanda Peña-Valencia, Diego Javier Jiménez. ✉e-mail: alexandre.rosado@kaust.edu.sa; a.reyes@uniandes.edu.co; diego.jimenezavella@kaust.edu.sa

