## [Transparent Peer Review File · Nature Communications]

Lignocellulose-mediated selection of potential halophilic PET-degrading enzymes from mangrove soil

Corresponding Author: Dr Diego Jimenez

Version 0:

Reviewer comments:

Reviewer #1

(Remarks to the Author)

This is an excellent and timely manuscript that provides valuable insight into the emergence of halophilic plastic-degrading enzymes and the ecological processes driving microbial adaptation under saline and desiccating conditions. The experimental design is solid and ecologically relevant, combining controlled microcosms with lignocellulose and polyethylene exposure to test selective pressures that mimic natural environments. The discovery of novel, potentially thermostable PETase-like enzymes highlights the metabolic versatility of extremophilic communities and will be of broad significance to environmental microbiology and biodegradation research.

The study is methodologically solid, with deep metagenomic sequencing (~65 million reads per sample; ~23.4 Gbp per treatment) enabling both genome-resolved and unbinned functional analyses. The use of the VEBA pipeline and the PlusPFP database is a major strength, as it supports standardized and comprehensive protein catalog generation across domains. However, the link between function and taxonomy remains unclear and, in its current form, not fully convincing. It appears that PETase-like sequences were identified from the unbinned protein catalog rather than within specific MAGs. If so, I would encourage the authors to clarify whether they explicitly searched for PETase genes within their MAGs and to discuss why these enzymes might not have been detected there (e.g., due to short contigs, plasmid localization, or limitations of binning). This clarification is essential to assess whether the absence of PETases in MAGs reflects biological reality or methodological constraints. Additionally, a phylogenetic analysis of the PETase-like sequences, placed alongside taxonomically validated reference PETases, would help verify their evolutionary relationships and possible genomic origins. Such an analysis could substantially strengthen the functional–taxonomic linkage and support the claim that these enzymes represent novel or halophilic variants.

The integration of 16S rRNA amplicon and shotgun metagenomic datasets is another strength, though the analytical relationship between them could be clarified. It would be helpful to specify whether 16S sequencing was used exclusively for community structure and diversity analyses, or if it also informed metagenomic taxonomic profiles. Additionally, a brief explanation of the Boruta algorithm (a random forest–based feature selection method) would improve methodological transparency.

Another valuable but unexplored aspect is the eukaryotic fraction. Fungal sequences appear to follow similar ecological patterns to bacterial degraders, particularly under lignocellulose exposure and reduced salinity. Given the well-documented role of fungi in producing cutinases, esterases, and oxidoreductases, a short discussion acknowledging this potential contribution would broaden the ecological scope of the study.

Overall, this is a high-quality and well-executed piece of research. The conclusions are supported by the data, and the methodology meets current standards in environmental metagenomics. The main areas for improvement are (i) a clearer explanation of how VEBA, PlusPFP, and MAG reconstruction were used in tandem; (ii) explicit acknowledgment of the limitations of the functional–taxonomic linkage; and (iii) a brief perspective on the fungal component. Addressing these points would further strengthen an already impressive and impactful contribution.

Reviewer #2

(Remarks to the Author)

Comments to Pena-Valencia et al.

The manuscript entitled "Lignocellulose-mediated selection of halophilic PET-degrading enzymes from mangrove soil" by Pena-Valencia et al. reported prokaryotic and fungal community perturbations in microcosm incubations induced by the addition of seawater, PET, and lignin. The authors used marker-gene amplicon and metagenome sequencing to characterize the community changes (particularly PET enzyme-containing species enrichment) in different treatments. I am not a PET expert and cannot judge the parts related to PET enzymes and structures. I don't know how big biotechnological application potentials these findings have. I apologize first if my criticisms below delay the publication of this manuscript. I find it lack focal points from a microbiologist's perspective. It covers many aspects, but lack deep insights on most of them. Sequencing data alone cannot decipher the complex communities and their responses to perturbations in mangrove soils. I have the following two suggestions for the authors to consider collecting some wet lab data to reinforce their findings. It is clear that some lineages became more abundant than others, which can be due to multiple reasons. Is it possible that they are just more resistant to the imposed unfavorable conditions relative to others? I can see some of the discussed organisms seems to be quite cosmopolitan and can be seen in many human-affected environments. Their abundance changes over time could be the key to really reveal the population dynamics (growth/decay) of each discussed organismal group.

A lot of PET enzyme findings were presented and discussed, based on the dry lab sequencing data. Are there any data to support that those organisms are really responding to the imposed PET/lignin additions in the incubations, like transcripts? Moreover, is it possible to measure the concentrations of the added PET/lignin or their degradation products over time to prove the activities of the bulk communities? Finally, is it possible to have some PET-enzyme assays to check if they have the assumed activities or not.

Version 1:

Reviewer comments:

Reviewer #1

(Remarks to the Author)

The authors have satisfactorily addressed my concerns.

Reviewer #2

(Remarks to the Author)

The authors had mostly addressed my questions and concerns.

Responses to the reviewer's comments are as described below (in blue):

Reviewer #1:

This is an excellent and timely manuscript that provides valuable insight into the emergence of halophilic plastic-degrading enzymes and the ecological processes driving microbial adaptation under saline and desiccating conditions. The experimental design is solid and ecologically relevant, combining controlled microcosms with lignocellulose and polyethylene exposure to test selective pressures that mimic natural environments. The discovery of novel, potentially thermostable PETase-like enzymes highlights the metabolic versatility of extremophilic communities and will be of broad significance to environmental microbiology and biodegradation research.

R/ Thank you for the positive assessment of our article. We are glad that you have found it solid, timely, and interesting.

The study is methodologically solid, with deep metagenomic sequencing (~65 million reads per sample; ~23.4 Gbp per treatment) enabling both genome-resolved and unbinned functional analyses. The use of the VEBA pipeline and the PlusPFP database is a major strength, as it supports standardized and comprehensive protein catalog generation across domains. However, the link between function and taxonomy remains unclear and, in its current form, not fully convincing. It appears that PETase-like sequences were identified from the unbinned protein catalog rather than within specific MAGs. If so, I would encourage the authors to clarify whether they explicitly searched for PETase genes within their MAGs and to discuss why these enzymes might not have been detected there (e.g., due to short contigs, plasmid localization, or limitations of binning). This clarification is essential to assess whether the absence of PETases in MAGs reflects biological reality or methodological constraints. Additionally, a phylogenetic analysis of the PETase-like sequences, placed alongside taxonomically validated reference PETases, would help verify their evolutionary relationships and possible genomic origins. Such an analysis could substantially strengthen the functional–taxonomic linkage and support the claim that these enzymes represent novel or halophilic variants.

R/ Indeed, the metagenomic information obtained from MAGs can be constrained by the binning process, where multiple low-quality bins, unbinned contigs, and short sequences are excluded in subsequent analyses (e.g., functional gene annotation). This becomes an important issue to be considered in environmental samples containing high levels of microbial diversity, which cause a low recovery of high-quality MAGs (see discussion in lines 378-391). Thus, to circumvent this issue, our search for predictive PET hydrolases (PETases) was performed in high-quality MAGs and in the full gene catalog obtained -via the VEBA pipeline- from binned and unbinned contigs

(see lines 262-266). As we mentioned in the text, MAGs provide a reasonable taxonomic resolution, sufficient to establish an accurate link between taxa and the detected enzymes (see lines 382-384). In our study, the majority of twelve putative PETases-like enzymes (except L8 and L11) were detected within unbinned contigs or low-quality bins and not within the set of high-quality MAGs. This suggests that some of these enzymes are likely within species present at low abundance in our samples, which limits their MAG recovery. In our study, this issue was surpassed by using both gene and genome-resolved metagenomics. We clarified this point in the revised text (see lines 406-411). In addition, taxonomy affiliation of binned and unbinned contigs was also carried out via the VEBA pipeline (information included in methods, see lines 601-602), this helps to clarify and support functional–taxonomic linkage.

Moreover, comprehensive phylogenetic analyses were performed in our study. For instance, in Fig. 6D, we built a phylogenetic tree using all putative PETases detected in MAGs and the full gene catalog from treatment L, in addition to thirteen prototypical and validated PET-active reference enzymes (such as LCC, *Is*PETase, and PHL7) deposited in the PAZy database (<https://onlinelibrary.wiley.com/doi/10.1002/prot.26325>). To complement this analysis, the same set of enzymes was used to build a phylogenetic tree using 3D structural features. Additionally, a complete phylogenetic tree was reconstructed using all PET-active enzymes available in the PAZy database alongside four putative PETases from *Marinobacter* (Mar1 and Mar2) and *Microbulbifier* (Mic1 and Mic2) (see Suppl. Fig. 2, lines 286-287). All these computational analyses allowed us to gain insights into the evolutionary relationship, novelty, and taxonomic origin of predictive PETases.

The integration of 16S rRNA amplicon and shotgun metagenomic datasets is another strength, though the analytical relationship between them could be clarified. It would be helpful to specify whether 16S sequencing was used exclusively for community structure and diversity analyses, or if it also informed metagenomic taxonomic profiles. Additionally, a brief explanation of the Boruta algorithm (a random forest–based feature selection method) would improve methodological transparency.

R/ For clarity, whole metagenomic sequencing was performed only at the end of the microcosm experiment (i.e., after 90 days of incubation). This decision was based on the information obtained via bacterial 16S rRNA gene amplicon sequencing, where drastic changes in the initial microbial communities' structure were observed (see Fig. 2). These changes were mostly associated with selection caused by desiccation and salinity. This information is now provided in the revised text (see lines 163-166).

The Boruta algorithm is a machine learning algorithm used for feature selection in highly dimensional datasets, including microbiome data (see <https://pmc.ncbi.nlm.nih.gov/articles/PMC6433899/>). The approach allows for the identification

of features that account for differences in microbial community composition or functional profiles across treatment groups. Briefly, the Boruta algorithm identifies important features (i.e., ASVs) by comparing the importance of real predictors to that of randomized “shadow” features. These shadow features are generated by permuting the original data across samples. In doing so, this permutation removes the real structure and the associations of treatments with response variables (i.e., ASVs). Importance values of the permuted data are generated through Random Forest and then compared to the real data. A feature is only considered important if its Random Forest importance score is consistently greater than that of the best-performing randomized variable. In other words, for an ASV to be important, it must do a better job of predicting the treatment group than the best-performing randomized value. This approach ensures that the selected features are more informative than what could be expected by random chance. Shadow feature comparison is well-suited for microbiome data (i.e., high-dimensional and sparse), where many ASVs may be weakly predictive due to random chance as opposed to real treatment associations.

The Boruta algorithm was implemented in R (see: <https://www.jstatsoft.org/article/view/v036i11/0>) and used to detect the features (i.e., ASVs) that explained variations between the treatment groups (i.e., seawater addition or lignocellulose addition). Briefly, bacterial ASVs indicative of treatment groups were modeled independently using random forest analysis with 1,000 trees, followed by feature selection using the Boruta algorithm. The abundance of significant ASVs (e.g., importance > 2) was then z-score transformed and visualized with heat maps built using ggplot2 (as is reported in <https://www.sciencedirect.com/science/article/pii/S0167779924002427#bbb0570>). This information is now included in the methods section (see lines 521-527).

Another valuable but unexplored aspect is the eukaryotic fraction. Fungal sequences appear to follow similar ecological patterns to bacterial degraders, particularly under lignocellulose exposure and reduced salinity. Given the well-documented role of fungi in producing cutinases, esterases, and oxidoreductases, a short discussion acknowledging this potential contribution would broaden the ecological scope of the study.

R/ We agree that fungal species can harbor potentially multiple biotechnologically interesting enzymes. However, in our study, bacterial-derived sequences dominated in the metagenome data (~ 95% in most of the samples) with a very low proportion of unassembled sequences (~ 0.2%) assigned to fungi via Kraken2. We did not consider exploring these low-abundant sequences and revised this information in the main text (see lines 391-395).

Overall, this is a high-quality and well-executed piece of research. The conclusions are supported by the data, and the methodology meets current standards in environmental metagenomics. The main areas for improvement are (i) a clearer explanation of how VEBA, PlusPFP, and MAG reconstruction were used in tandem; (ii) explicit acknowledgment of the limitations of the

functional–taxonomic linkage; and (iii) a brief perspective on the fungal component. Addressing these points would further strengthen an already impressive and impactful contribution.

R/ Thanks for your comments. First, Kraken/Bracken-PlusPFP was used to taxonomically classify unassembled sequences, the VEBA pipeline was used to develop a full gene catalog from binned and unbinned contigs, and MAG reconstruction was carried out to obtain taxonomic and functional insights into the most abundant microorganisms within the soil microcosms. Second, the taxonomic origin of putative PETases was determined by the taxonomic affiliation of binned or unbinned contigs (via VEBA) where genes were detected. This taxonomic assignment was supported by amino acid sequence analyses and 3D structure comparison along prototypical PET-active enzymes. Third, the fungal component and its potential were mentioned in the revised text.

Reviewer #2:

Comments to Pena-Valencia et al.

The manuscript entitled “Lignocellulose-mediated selection of halophilic PET-degrading enzymes from mangrove soil” by Pena-Valencia et al. reported prokaryotic and fungal community perturbations in microcosm incubations induced by the addition of seawater, PET, and lignin. The authors used marker-gene amplicon and metagenome sequencing to characterize the community changes (particularly PET enzyme-containing species enrichment) in different treatments. I am not a PET expert and cannot judge the parts related to PET enzymes and structures. I don’t know how big biotechnological application potentials these findings have. I apologize first if my criticisms below delay the publication of this manuscript.

R/ Thank you for all the comments provided. For clarity, our study is located between the microbial ecology-enzyme bioprospecting/discovery zone. It provides (i) diversity- and genomic-based insights that advance our understanding of mangrove soil microbiomes under changing environmental conditions, and (ii) a conceptual and methodological framework to improve the selection of putative PET-degrading enzymes from environmental microbiomes. Importantly, this study does not aim to demonstrate or validate the industrial applicability of the identified enzymes. Rather, based on amino acid sequence analyses, predicted 3D structures, and AI-assisted predictions, we reported enzymatic features (e.g., halotolerance and thermostability) that suggest these candidate PETases may be promising targets for downstream experimental characterization, benchmarking, and evaluation under industrially relevant conditions.

I find it lack focal points from a microbiologist’s perspective. It covers many aspects, but lack deep insights on most of them. Sequencing data alone cannot decipher the complex communities and their responses to perturbations in mangrove soils. I have the following two suggestions for the authors to consider collecting some wet lab data to reinforce their findings.

It is clear that some lineages became more abundant than others, which can be due to multiple reasons. Is it possible that they are just more resistant to the imposed unfavorable conditions relative to others? I can see some of the discussed organisms seems to be quite cosmopolitan and can be seen in many human-affected environments. Their abundance changes over time could be the key to really reveal the population dynamics (growth/decay) of each discussed organismal group.

R/ Thank you for this positive feedback provided and to highlight potential issues under the view of microbiologists. As specifically discussed in our article, the absence of seawater, the inclusion of lignocellulose, and the often high salinity conditions caused by desiccation were the major drivers that selectively impacted the microbial community structure in our microcosm system (see lines 341-354). These factors operating over time resulted in the selection of resistant microorganisms (e.g., spore-forming, halotolerant, and lignocellulolytic ones) after 90 days of incubation in treatment L, increasing their relative abundance compared with others. As mentioned by the reviewer, some of these selected taxa belong to cosmopolitan halophilic groups (e.g., *Bacillacea*), although we have also found specialized taxa thriving under these conditions in our microcosm system (e.g., *Micromonosporaceae*). Probably, extensive sampling over time can provide information about population dynamics; however, this goal was not initially considered in our experimental design. Based on your comment, we acknowledged in the revised text that alternative techniques such as metatranscriptomics can provide insight in microbial activity and also population dynamics after perturbation events (see lines 371-376).

A lot of PET enzyme findings were presented and discussed, based on the dry lab sequencing data. Are there any data to support that those organisms are really responding to the imposed PET/lignin additions in the incubations, like transcripts? Moreover, is it possible to measure the concentrations of the added PET/lignin or their degradation products over time to prove the activities of the bulk communities? Finally, is it possible to have some PET-enzyme assays to check if they have the assumed activities or not.

R/ We acknowledge the importance of the points raised by the reviewer. We consider that the metagenomic-derived (dry-lab sequencing data) results are sufficiently robust and support the claims and arguments developed in the main text.

In our study, predictive PETases were enriched in the metagenomes from lignocellulose-added microcosm (L treatment) compared to controls (Fig. 6A). Moreover, specific bacterial taxa (e.g., *Micromonospora*, *Streptomyces*, and *Microbulbifer*) that hold these putative PETases were primarily responding to the input of rice husk, but not to the input of PET. In this regard, these taxa are known for their capacity to degrade plant biomass, and they can be metabolically active within the microcosms. Moreover, rice husk material was almost completely consumed (see photos in Fig. 1B), and the remaining plant biomass (after 90 days of incubation) would be the hardest

fraction to be degraded (i.e., lignin). We have mentioned this in the revised text (see lines 336-338). Likewise, the amendment with PET particles did not result in detectable effects on the microbial communities compared to controls (Fig. 2B). In addition, depolymerization of PET (and release and consumption of PET-derived monomers) is unlikely to occur in the established microcosm conditions. As we mentioned in the text, PET-added microcosms did not favor the selection of PET-degrading taxa, likely due to the recalcitrance of PET to depolymerization, the availability of alternative carbon sources, or harsh environmental conditions that constrain PET degradation (see lines 397-400).

In the revised text, we also acknowledge that metatranscriptomic data can be used to validate the metabolic activity of the enriched taxa at the time of sampling (see lines 371-376). However, our dual-approach (i.e., gene-centric and genome-resolved metagenomics) was useful to explore the metabolic potential of microbial taxa within complex community contexts. Moreover, as the microcosm experiment was carried out in 2021, metatranscriptome sequencing, analysis of transcripts, and measurement of lignin content from the same flasks is now technically impossible. We carefully acknowledge some of these limitations in our study, see lines 371-376.

Last, the predictive enzymatic activity of the twelve putative PETases is strongly supported by the presence of catalytic features, including the catalytic triad, disulfide bonds, the M5 motif, as well as phylogenetic placement, amino acid sequence similarity, and predicted 3D structural similarity to previously characterized PET-active enzymes. Comprehensive wet-lab validation of enzymatic activity for these twelve candidates PETases would involve complex experimental processes, would be time-consuming, and can face multiple experimental bottlenecks (e.g., heterologous expression, protein solubility issues, and enzymatic tests). In addition, it might fall outside the scope of the present study, as detailed enzymatic activity characterization, benchmarking, and assessments of industrial applicability were not initially considered in our main aims (see aims in lines 113-119). Nevertheless, we explicitly acknowledge this next step and highlight it as a future research direction in the revised text (see conclusion section, lines 473-476).